# A complex aerosol transport event over Europe during the 2017 Storm Ophelia in CAMS forecast systems: analysis and evaluation

Dimitris Akritidis[1], Eleni Katragkou[1], Aristeidis K. Georgoulias[1], Prodromos Zanis[1], Stergios Kartsios[1], Johannes Flemming[2], Antje Inness[2], John Douros[3], and Henk Eskes[3]

[1]Department of Meteorology and Climatology, School of Geology, Aristotle University of Thessaloniki, Thessaloniki, Greece
[2]European Centre for Medium-Range Weather Forecasts (ECMWF), Reading, UK
[3]Royal Netherlands Meteorological Institute (KNMI), De Bilt, the Netherlands

**Correspondence:** D. Akritidis (dakritid@geo.auth.gr)

**Abstract.** In mid-October 2017 Storm Ophelia crossed over western coastal Europe, inducing the combined transport of Saharan dust and Iberian biomass burning aerosols over several European areas. In this study we assess the performance of the Copernicus Atmosphere Monitoring Service (CAMS) forecast systems during this complex aerosol transport event, and the potential benefits that data assimilation and regional models could bring. To this end, CAMS global and regional forecast data are analyzed and compared against observations from passive (MODIS: Moderate resolution Imaging Spectroradiometer aboard Terra and Aqua) and active (CALIOP/CALIPSO: Cloud-Aerosol Lidar with Orthogonal Polarization aboard Cloud-Aerosol Lidar and Infrared Pathfinder Satellite Observations) satellite sensors, and ground-based measurements (EMEP: European Monitoring and Evaluation Programme). The analysis of CAMS global forecast indicates that dust and smoke aerosols, discretely located on the warm and cold front of Ophelia, respectively, are affecting the aerosol atmospheric composition over Europe during the passage of the Storm. The observed MODIS Aerosol Optical Depth (AOD) values are satisfactorily reproduced by CAMS global forecast system, with a correlation coefficient of 0.77 and a fractional gross error (FGE) of 0.4. The comparison with a CAMS global control simulation not including data assimilation, indicates a significant improvement in the bias due to data assimilation implementation, as the FGE decreases by 32%. The qualitative evaluation of the IFS (Integrated Forecast System) dominant aerosol type and location against the CALIPSO observations overall reveals a good agreement. Regarding the footprint on air quality, both CAMS global and regional forecast systems are generally able to reproduce the observed signal of increase in surface particulate matter concentrations. The regional component performs better in terms of bias and temporal variability, with the correlation deteriorating over forecast time. Yet, both products exhibit inconsistencies on the quantitative and temporal representation of the observed surface particulate matter enhancements, stressing the need for further development of the air quality forecast systems, for even more accurate and timely support of citizens and policy-makers.

## 1 Introduction

Atmospheric aerosols play a prominent role in atmospheric composition, climate and human health (Pöschl, 2005; IPCC, 2013). Given the broad variety of their natural and anthropogenic sources, their relatively short lifetime, and their different formation mechanisms, aerosols exhibit highly variable spatiotemporal distribution around the globe (Putaud et al., 2010; Boucher, 2015).

Over Europe, apart from local emissions, particulate matter quantities are also determined by atmospheric transport through mesoscale weather systems (Ansmann et al., 2003; Kallos et al., 2007; Pey et al., 2013), occasionaly implying significant implications for air quality and public health (Stafoggia et al., 2016). Consequently, the operational forecast of atmospheric composition is essential in the direction of supporting social and health policy-makers.

Aerosols interact with solar radiation directly through scattering and absorption (Haywood and Boucher, 2000), and indirectly by modifying the micro- and macro-physical properties of clouds (Lohmann and Feichter, 2005), modulating the energy balance in the earth-atmosphere system, and by extension the climate. More specifically, they may act as cloud condesation nuclei (CCN) and ice nuclei (IN) increasing the cloud albedo under constant water paths (Twomey, 1974), as well as cloud lifetime and coverage suppressing the precipitation formation (Albrecht, 1989). In addition, the presence of absorbing aerosols

alters the thermodynamic state of the atmosphere (semi-direct effect) affecting clouds and precipitation in various ways (Koch and Del Genio, 2010).

Exposure to particulate matter air pollution is well known to have significant implications to respiratory- and cardiovascular-based mortality (Russell and Brunekreef, 2009; Lelieveld et al., 2015; Burnett et al., 2018). Fine particulate matter with a diameter less than 2.5 $\mu$m (PM$_{2.5}$) has more profound effects on human health compared to coarse particulate matter with a

diameter less than 10 $\mu$m (PM$_{10}$), as smaller particles can penetrate more efficiently into the lungs and the indoor environments, while they can also be transported over greater distances given their smaller mass and therefore their longer lifetime (Pope and Dockery, 2006). Recently, Lelieveld et al. (2019) based on new hazard ratio functions and ambient air pollution exposure data estimated that air pollution reduces the mean life expectancy by more than 2 years in Europe. Moreover, aerosols can pose significant hazards for aviation, occasionally resulting in flight delays and cancellations, as intense episoding dust and sea

salt events can alter visibilty (Gertisser, 2010; Tighe, 2015), while the presence of dust and volcanic particles may potentially cause aircraft engine damage (Gertisser, 2010). The impact of aerosols on photovoltaic generation is mainly due to reduction of surface solar radiation through scattering and absorption (Li et al., 2017), and deposition of dust on solar panels reducing their energy production potential (Beattie et al., 2012).

North African dust is the dominant source of mineral dust in the atmosphere (Ginoux et al., 2012), affecting the air quality

of Europe from the Mediterranean (Querol et al., 2009; Georgoulias et al., 2016) up to Scandinavia (Ansmann et al., 2003). The main transport pathways of Saharan dust towards Europe are either passing through the Mediterranean by northward flow associated with the presence of cyclones/anticyclones westward/eastward of a Saharan dust source, or via the Atlantic route including a westward transport of Saharan dust by trade winds over the Atlantic followed by northward and then eastward flow towards Europe (Israelevich et al., 2012). Over Europe, biomass burning emissions from wildfires are a present air pollution

risk, likely to increase in a changing climate (Knorr et al., 2017; Turco et al., 2017). The regional scale (Amiridis et al., 2009) and intercontinental (Markowicz et al., 2016) transport of smoke over Europe may affect the atmospheric composition and the local air quality (Sciare et al., 2008).

During the recent years various global and regional integrated forecast systems incorporating the online coupling between meteorology and atmospheric composition have been developed to support research, forecast and policy activities (Baklanov

et al., 2014). Copernicus (Copernicus, 2020) is the European sytem for monitoring Earth coordinated by the European Union.

The Copernicus Atmosphere Monitoring Service (CAMS) (CAMS, 2020a) is one of the six services that Copernicus provides, using a comprehensive global assimilation and forecasting system to assess the state and composition of the atmosphere on a daily basis. It incorporates information from models and observations, providing daily 5-day forecasts of atmospheric composition fields, such as chemically reactive gases and aerosols (Flemming et al., 2015; Inness et al., 2015). The CAMS global modeling system is also used to provide the boundary conditions for the CAMS ensemble of regional air quality models, which produce 4-day forecasts of European air quality at higher horizontal resolution. To increase confidence of operational use, unravel limitations and further improve the CAMS forecast systems, analysis and evaluation of its performance during complex and challenging situations is deemed necessary.

On the morning of October 16$^{th}$ 2017 a red sky phenomenon was reported on several sites of the United Kingdom (UK) (Telegraph, 2017), as a result of the high aerosol loadings in the overlying atmosphere (Harrison et al., 2018; Osborne et al., 2019), while similar reports on the morning of October 17$^{th}$ were given for the Netherlands as well (NLTIMES, 2017). These high aerosol loadings were the result of transport of desert dust from Northern Africa and smoke from Iberian wildfires. This study aims to analyze and evaluate the performance of CAMS global and regional forecast systems during this challenging combined dust and biomass burning transport event over western Europe induced from the passage of Storm Ophelia in mid-October 2017. To this end, CAMS global and regional day-1 forecast data are used, along with ground-based observations (EMEP: European Monitoring and Evaluation Programme), and observations from passive (MODIS: Moderate resolution Imaging Spectroradiometer aboard Terra and Aqua) and active (CALIOP/CALIPSO: Cloud-Aerosol Lidar with Orthogonal Polarization/Cloud-Aerosol Lidar and Infrared Pathfinder Satellite Observations) satellite remote sensors. Furthermore, the impact of data assimilation on the CAMS global aerosol burden representation is also explored using a control simulation without the use of data assimilation. Finally, to explore the forecast capability over time, CAMS global and regional day-2 and day-3 forecasts of PM$_{10}$ and PM$_{2.5}$ concentrations are also analyzed. This paper builds upon the work of Akritidis et al. (2018), assessing the perfomance of CAMS forecast products, the potential benefits of data assimilation in CAMS global system and the added value that CAMS regional models could bring, during a complex aerosol transport event. The structure of the paper is as follows. Section 2 presents the CAMS forecast systems and the observational data that are used to validate their perfomance. Section 3 shows the analysis and evaluation results, and Section 4 summarises the key findings.

## 2 CAMS forecast systems, validation data and metrics

### 2.1 CAMS global forecast system

One of the core CAMS products is the 5-day global chemical weather forecast, based on the European Centre for Medium-Range Weather Forecasts (ECMWF) Integrated Forecasting System (IFS). In October 2017, the forecast data were products of the IFS model cycle 43r3 having a horizontal resolution of about 40 km and 60 vertical levels reaching up to 0.1 hPa. The IFS modeling system uses an extended version of the Carbon Bond 2005 (CB05) chemical mechanism (Flemming et al., 2015) and the Morcrette et al. (2009) aerosol parameterization. Several chemical species, including ozone, nitrogen dioxide, carbon monoxide, sulfur dioxide are assimilated using products of several satellite missions (Inness et al., 2015, 2019b, and

references therein). Moreover, the IFS assimilates total Aerosol Optical Depth (AOD) retrievals from the MODIS instrument on NASA's Aqua and Terra satellite, as well as PMAp (Polare Multi Sensor Aerosol product) aerosol retrievals over sea, a combined GOME-2/AVHRR product produced by Eumetsat (EUMETSAT, 2020), and subsequently updates the individual aerosol components based on their fractional contribution to the total aerosol mass (Benedetti et al., 2009; Georgoulias et al., 2018). Five types of aerosols are included, namely sea salt, dust, hydrophylic and hydrophobic organic matter and black carbon, and sulfate. The first two aerosol types are provided in three radius size bins: 0.03-0.5, 0.5-5 and 5-20 $\mu$m and 0.03–0.55, 0.55–0.9 and 0.9–20 $\mu$m, respectively. The different IFS aerosol types are treated as externally mixed (Inness et al., 2019a).

Here we use IFS day-1 (referred here as IFS) forecasts (initiated at 00:00Z) of temperature, mean sea level pressure (mslp), omega vertical velocity, divergence, u and v wind components, mass mixing ratio of 11 aerosol variables, total AOD at 550 nm (AOD$_{550}$ or just AOD), sea salt AOD, dust AOD, organic matter AOD, black carbon AOD, sulfate AOD and carbon monoxide (CO) mass mixing ratio. Also used are day-1 to day-3 forecast data of PM$_{10}$ and PM$_{2.5}$ surface concentrations, which are derived according to the following formula:

$$PM_{10} = \rho \cdot (\ SS1\ /\ 4.3\ +\ SS2\ /\ 4.3\ +\ DD1\ +\ DD2\ +\ 0.4 \cdot DD3\ +\ OM1\ +\ OM2 \\ +\ SU1\ +\ BC1\ +\ BC2\ ) \tag{1}$$

$$PM_{2.5} = \rho \cdot (\ SS1\ /\ 4.3\ +\ 0.5 \cdot SS2\ /\ 4.3\ +\ DD1\ +\ DD2\ +\ 0.7 \cdot OM1\ +\ 0.7 \cdot OM2 \\ +\ O.7 \cdot SU1\ +\ BC1\ +\ BC2\ ) \tag{2}$$

where $\rho$ the air density, SS1 the sea salt radius size bin 1 (0.03-0.5 $\mu$m), SS2 the sea salt radius size bin 2 (0.5-5 $\mu$m), DD1 the desert dust radius size bin 1 (0.03–0.55 $\mu$m), DD2 the desert dust radius size bin 2 (0.55–0.9 $\mu$m), DD3 the desert dust radius size bin 3 (0.9–20 $\mu$m), OM1 the hydrophobic organic matter, OM2 the hydrophilic organic matter, BC1 the hydrophobic black carbon, BC2 the hydrophilic black carbon and SU1 the aerosol sulfate (ECMWF, 2020). A detailed description and evaluation of the aerosol scheme used in IFS can be found in Rémy et al. (2019). To unravel the impact of chemical data assimilation on aerosol burden representation during the examined event, an additional IFS control run without data assimilation (no DA) is also used for intercomparison.

## 2.2 CAMS regional forecast system

The CAMS regional forecast system provides the ensemble median (RegEns) and members of the European-scale air quality forecasts for every hour up to four days in advance. The products have a 0.1° horizontal resolution and are based on state-of-the-art numerical air quality models developed in Europe: CHIMERE from INERIS (National Institute for Industrial Environment and Risks) (Menut et al., 2014), DEHM from Aarhus University (Christensen, 1997), EMEP from MET-Norway (Simpson et al., 2012), EURAD-IM from the University of Cologne (Memmesheimer et al., 2004), GEM-AQ from IEP-NRI (Institute of Environmental Protection, Polism Ministry of Environment) (Kaminski et al., 2008), LOTOS-EUROS from KNMI (Royal Netherlands Meteorological Institute) and TNO (Netherlands Organisation for Applied Scientific Research) (Manders et al., 2017), MATCH from SMHI (Swedish Meteorological and Hydrological Institute) (Robertson et al., 1999), MOCAGE from

Météo-France (Guth et al., 2016) and SILAM from FMI (Finnish Meteorological Institute) (Sofiev et al., 2015). More details on the regional air quality systems can be found at CAMS (2020c). The global ECMWF IFS forecasts provide the meteorological and the chemical boundary forcing to the regional model suite and CAMS provides the emissions. Several CAMS regional models assimilate $PM_{10}$ and $PM_{2.5}$ surface observations from various stations of the EEA's (European Environment Agency)

Air Quality e-reporting database, but not satellite aerosol products. More specifically, during the period of interest (October 2017), CHIMERE and EURAD were assimilating both $PM_{10}$ and $PM_{2.5}$, MOCAGE only $PM_{10}$ and, finally, SILAM and MATCH were assimilating only $PM_{2.5}$. $PM_{10}$ and $PM_{2.5}$ concentrations in the regional models are calculated using simulated aerosol fields specific to each regional model. All models are validated operationally (CAMS, 2020b) and a posteriori (CAMS, 2020a), and operational verification results are available daily for six chemical species (ozone, nitrogen dioxide, sulfur dioxide,

carbon monoxide, $PM_{10}$ and $PM_{2.5}$). It has to be noted that for the examined period (October 2017) DEHM and GEM-AQ models are not included in RegEns, as they were recently added in the CAMS regional forecast system. RegEns day-1 (referred here as RegEns), day-2 and day-3 forecast data (initiated at 00:00Z) of $PM_{10}$ and $PM_{2.5}$ surface concentrations are used, to analyze and evaluate the performance of CAMS regional forecast system during the examined event.

## 2.3 Validation data

To evaluate the CAMS forecast systems ability to capture the spatial and temporal distribution of aerosols in the atmosphere, $AOD_{550}$ data from the two MODIS sensors aboard EOS Terra (equator crossing time 10:30 LT (Local Time)) and Aqua (equator crossing time 13:30 LT) satellites are used along with aerosol subtype data from CALIOP/CALIPSO (equator crossing time ~13:30 LT). MODIS is a 36-band imaging radiometer with a viewing swath of 2330 km offering almost daily global coverage (Salomonson et al., 1989). In this work, level-2 data with a resolution of 10 x 10 $km^2$ at nadir

from the MODIS Collection 6.1 combined Dark Target algorithm (DT) and Deep Blue algorithm (DB) scientific dataset AOD_550_Dark_Target_Deep_Blue_Combined are processed. There are two different DT algorithms, one for retrievals over land (Kaufman et al., 1997; Remer et al., 2005; Levy et al., 2013) and one for water surfaces (Tanré et al., 1997; Remer et al., 2005; Levy et al., 2013), while DB currently delivers retrievals over all land types (Hsu et al., 2013; Sayer et al., 2013, 2014, 2015) despite the fact that originally was developed for bright land surfaces only (Hsu et al., 2004). For the quantitative

evaluation of the CAMS forecasts the MODIS/Terra and Aqua data were merged and brought to the CAMS native grid.

CALIOP is a spaceborne lidar instrument (Hunt et al., 2009) providing profiles of aerosol and cloud-related properties (Winker et al., 2009) within the first 30 km of the atmosphere by measuring the backscatter signals and the polarization of the backscattered light. The CALIPSO algorithm identifies distinct atmospheric layers (clean air, aerosols, clouds, surface, etc.) (Vaughan et al., 2009; Kim et al., 2018) and attributes a specific aerosol subtype (marine, dust, polluted continental/smoke,

clean continental, polluted dust, elevated smoke, dusty marine, polar stratospheric cloud aerosols, volcanic ash, sulfate/other) to each of them (Omar et al., 2009). In this work, data from the CALIOP/CALIPSO version 4.20 level-2 product (Kim et al., 2018) with a horizontal resolution of 5 km and a vertical resolution of 30 m (for heights below ~20 km) are processed.

To assess CAMS forecast systems' performance in reproducing the impacts on air quality during the Ophelia passage ground-based observations are used. Measurements of $PM_{10}$ and $PM_{2.5}$ surface concentrations from 8 rural background stations (see

Table 1 for details) are obtained from the EMEP (Tørseth et al., 2012) network through the EBAS database (EBAS, 2020) for the time period from 10 October 2017 to 20 October 2017. The stations are located over western Europe and away from the dust and biomass burning sources, lie across the plumes of high AOD loadings, exhibiting significant increases in $PM_{10}$ and $PM_{2.5}$ surface concentrations during the examined event. Data are provided with 1-hour temporal resolution, yet a 3-hour resolution is used for direct comparison with the IFS data. It has to be noted that from the examined stations only GB0043R and GB0048R for $PM_{10}$, and GB0048R for $PM_{2.5}$, are listed as assimilation stations for the CAMS regional models.

## 2.4 Statistical metrics

To evaluate the performance of CAMS forecast systems with respect to observational data, the following statistical metrics are used.

a) The temporal correlation of CAMS models with observations is assessed with the Pearson correlation coefficient (R) that measures the strength of their linear association, ranging between + 1 and -1:

$$R = \frac{N \sum M_i O_i - \sum M_i \sum O_i}{\sqrt{N \sum M_i^2 - (\sum M_i)^2} \sqrt{N \sum O_i^2 - (\sum O_i)^2}} \tag{3}$$

where $M_i$ and $O_i$ are the modeled and observed values, respectively, and N is the number of the sample.

b) The model error is estimated using the fractional gross error (FGE) which ranges between 0 and 2, and behaves symmetrically with respect to under- and overestimation:

$$FGE = \frac{2}{N} \sum_i^N \left| \frac{M_i - O_i}{M_i + O_i} \right| \tag{4}$$

where $M_i$ and $O_i$ are the modeled and observed values, respectively, and N is the number of the sample.

## 3 Results

### 3.1 Storm Ophelia and transport pathways

Hurricane Ophelia occuring in October 2017 was an exceptional low pressure system, as it had unique characteristics as an Atlantic hurricane, it caused the death of three people and extended damages during its passage over Ireland (BBC, 2017), and indirectly affected the air quality and the atmospheric composition over several western European areas. Ophelia initiated as a non-tropical low pressure system over the Atlantic at the southwest of the Azores in early October 2017, and didn't have the fate as that of a common Atlantic tropical storm that it heads to the west (Stewart, 2018). Instead, Ophelia marched northeastwards initially reaching coastal Portugal, and subsequently loosing its tropical characteristics downgrading to an extratropical cyclone (ex-hurricane) it followed a northern path towards the Great Britain, becoming the easternmost major Atlantic hurricane ever recorded (Stewart, 2018). The aforementioned route of

Ophelia resulted from the guidance of an upper level trough located over the Atlantic (Stewart, 2018; Rantanen et al., 2020). Figure 1 presents the prevailing synoptic conditions in the middle troposphere (500 hPa), as seen from IFS, during the period from 12Z on 14 October 2017 to 00Z on 16 October 2017. A mid-latitude trough dominates over the central Atlantic, with the surface low pressure system located ahead of the trough and to the east of it, being driven from the northeasterly oriented wind flow (Fig. 1a and b). On 12Z of October $15^{th}$ and as the trough axis turns negatively tilted the flow becomes mostly southerly (Fig. 1c) dragging Ophelia towards the Ireland (Fig. 1d). Although after 12Z of October $14^{th}$ Ophelia moved over sea surface temperatures (SSTs) of about 25°C which in general do not support hurricane intensification (Stewart, 2018), the alternative fuel for Ophelia to maintain its strength was found in the enhanced upper level divergence on the eastern flank of the trough (not shown). This resulted in convection reinforcement, as revealed from the enhanced upward vertical velocity values at 500 hPa exceeding 2 Pa/s (Fig. 1).

From 00Z of October $13^{th}$ and onwards, Saharan dust being transported through the trade winds over the west coast of Africa, is exposed to the southerlies on the east side of Ophelia system, setting up a dust outbreak that gradually moves to the north. On 12Z of October $14^{th}$ a plume of high IFS dust mass mixing ratio exceeding 80 $\mu$g/kg is found west of the Iberian Peninsula up to approximately 700 hPa, as depicted in Figs. 2a and b. During the next 24 hours and as the surface low pressure system is traveling further north (at $\sim 40°$N on 12Z of October $15^{th}$), the dust particles entrained from the meridional flow at the east of Ophelias' periphery form an extended plume of high dust loadings reaching the southeast coast of UK (Figs. 2c, d, e and f). Throughout the following 12 hours the plume expands to the north and east passing over northwestern France and southern UK (Figs. 2g and h). On 00Z of October $16^{th}$ the Ophelia storm merges with the frontal system as can be seen in Fig. 2g and Fig. 3a, while smoke from wildfires that have been burning across the Iberian Peninsula is evident at 850 hPa, as high CO volume mixing ratio (> 200 ppbv), uplifted through the Warm Conveyor Belt (Osborne et al., 2019) on the southeast part of Ophelia where upward motion dominates (Fig. 2g and Fig.S1 in the Supplement). From this moment and until 18Z of October $16^{th}$, dust and smoke particles gather on the warm and cold front of Ophelia's warm sector, respectively, transported over western Europe through the frontal system (Figs. 3c, e, and g). This distinct discretization of dust and biomass burning within the warm sector of Ophelia is well illustrated by the 3-D fields of dust and CO mixing ratio exceeding 80 $\mu$g/kg and 200 ppbv respectively, resembling the shape of the warm and cold front (Figs. 3d, f, and h). Noteworthy is the uplift of smoke over northwest Iberian Peninsula from 18Z on 16 October to 06Z on 17 October (Figs. 2l), due to the profound upward motions induced from strong lower convergence and upper divergence (Fig.S1 in the Supplement) at the eastern flank of the upper level trough (not show). This elevated smoke plume is drifted during the next 24 hours from the northeasterly flow towards UK (Figs. 2n and p). The Ophelia storm starts dissipating from 00Z of October $17^{th}$ onwards fading away over Norway on 00Z of October $18^{th}$. A graphical representation of the IFS dust and CO transport along with the synoptic evolution during the passage of Ophelia (from 00Z of October $12^{th}$ 2017 to 21Z of October $20^{th}$ 2017) is provided as animation in the Supplement.

### 3.2 Aerosol atmospheric composition

Here we explore the impact of Ophelia's passage on the aerosol atmospheric composition over the broader European region, using CAMS global forecast data and satellite observations. The MODIS/Terra and Aqua $AOD_{550}$ spatial distribution for the time period from 14 October 2017 to 18 October 2017 is presented in Figure 4 (left column). It should be noted that these data don't give an independent validation as they are also assimilated in the IFS. High aerosol loadings (> 0.4) are observed at the west of the Iberian Peninsula at 14 October (Fig. 4a), while one day later the aerosol plume is extended further north and east over France, English Channel and Celtic Sea (Fig. 4d). On 16 and 17 October, high $AOD_{550}$ values are seen over North and Baltic Sea, France, Belgium, the Netherlands and Germany (Fig. 4g and j), while on 18 October the plume is found over Germany and Poland (Fig. 4m). The observed buildup of high AOD loadings over Europe is in line with the aforementioned description of the Ophelia storm and the associated transport pathways. To evaluate the CAMS global $AOD_{550}$ forecast product during this period, the respective $AOD_{550}$ fields are also shown in Fig. 4 (middle column). Overall, there is a good agreement between the CAMS global forecast product and the satellite observations, at least in qualitative terms, as the forecasted high AOD loadings resemble that of observations. To assess the impact of AOD data assimilation on IFS AOD forecast, the differences of $AOD_{550}$ with and without the use of data assimilation are also shown in Fig. 4 (right column), revealing that data assimilation boosts AOD values near the examined AOD plumes, while near the dust sources over Africa it mostly suppresses them. In addition, to quantitatively validate the CAMS global AOD forecast, a comparison of spatially and temporally collocated $AOD_{550}$ data between IFS (with and without the use of data assimilation) and MODIS/Terra and Aqua is performed for two box regions; the whole domain appearing in Fig. 4 and a smaller domain in northwestern Europe appearing in Fig. 9d. As depicted from the scatter plots of Figs. 5a and b for the large domain, the use of data assimilation improves the CAMS global forecast performance increasing the correlation (from 0.72 to 0.77) and reducing the error (FGE from 0.59 to 0.4) with respect to the control simulation (no DA). For a more "event-based" evaluation of CAMS global AOD forecast, the respective scatter plots for a small domain in northwestern Europe where the AOD plume is found are shown in Figs. 5c and d, indicating less agreement with the satellite observations and a slight improvement due to data assimilation implementation (correlation increase from 0.54 to 0.56, and FGE decreases from 0.48 to 0.44). Overall, the CAMS global AOD forecast product tends to overestimate for low observed AOD values and vice versa, a behavior that was also observed in a previous version of the ECMWF global atmospheric composition reanalysis dataset (MACC (Monitoring Atmospheric Composition and Climate); see Georgoulias et al. (2018) for details).

To identify the dominant IFS aerosol types within the high AOD plume, the modeled percentage (%) contribution of each aerosol type to the total AOD, as shown in Fig. 4, is presented in Fig. 6 for the same period. On 12Z of October $14^{th}$, the dominant source of the AOD loading found at the west of the Iberian Peninsula is mineral dust by 60-70%, while the dust contribution over UK is up to 20%, as depicted from Fig. 6a. One day later and as the AOD plume moves to the north and east, dust continues to dominate with percentage contribution of up to 45% over UK (Fig. 6f). On 12Z of October $16^{th}$, the dust AOD contribution over UK ranges with a percentage between 20-35%, while over the North Sea and

the Netherlands it reaches values up to 65% (Fig. 6k). The biomass burning emission trails originating from northwest Iberian Peninsula are also a significant input for AOD, with black carbon and organic matter aerosols contributing up to 20% and 40% over UK, respectively (Figs. 6m and n). Such black carbon contributions are considered high, being similar to climatological contributions over global fire hot spots (e.g. summertime central southern Africa; Penning de Vries et al. (2015)). The elongated plume of high AOD values on 12Z of October $17^{th}$ as seen from both the satellite observations and the CAMS global forecast (Figs. 4j and k), consists of up to ∼50% of organic matter aerosols (Fig. 6s), up to ∼25% of black carbon aerosols (Fig. 6r) and up to ∼20% of dust aerosols (Fig. 6p), while over the Baltic States the major contributor is mineral dust with ∼50%. This is consistent with the transport pathways of dust and biomass burning aerosols shown in Fig. 2, indicating that fire-originated aerosols dominate over northern coastal Europe, while dust aerosols remnants from Ophelia's warm front passage prevail over the Baltic States. A similar situation also occurs one day later as depicted from Figs. 6u, w and x. It is worth noting, that during the examined period sea salt aerosols are dominating over the Atlantic Sea, following the evolution of Storm Ophelia, as they are produced and transported due to the intense winds near Ophelia (Figs. 6b, g, l, q, v).

To assess the performance of CAMS global forecast in representing the vertical distribution of aerosol types during the passage of Ophelia, CALIPSO aerosol subtype satellite data were also utilized. Two CALIPSO tracks that crossed the aerosol plume during the examined period were selected for 15/10/2017 and 17/10/2017 with an overpass time around 13:00Z and 12:45Z, respectively (tracks are shown on the top left of Figs. 7a and c). For the selected CALIPSO tracks, spatially (horizontally and vertically) and temporally (at 12Z) collocated IFS aerosol mass mixing ratio data were extracted. Subsequently, the dominant (in terms of mass mixing ratio) aerosol type was identified, namely, sea salt, dust, carbonaceous aerosols (organic matter and black carbon) and sulfates, following Georgoulias et al. (2020). A dominant aerosol type was set only when the highest mass mixing ratio from the four subtypes exceeded 1 $\mu$g/kg. Figure 7 presents the vertical cross sections of CALIOP/CALIPSO aerosol subtypes and IFS dominant aerosol type along the two selected CALIPSO tracks. On 12Z of October $15^{th}$, a distinct presence of dust and polluted dust aerosols is observed between 44°N and 57°N, with polluted dust being confined within the first ∼2 km. Moreover, polluted continental/smoke aerosols are also detected over the UK (>50°N) up to approximately 1 km. Further south and over the Mediterranean Sea (36°N-43°N), sea salt and dusty marine aerosols are mostly dominating indicating a dust cross over the Mediterranean, while southern than 36°N, over North Africa, polluted dust is observed (Fig. 7a). The respective vertical cross section of IFS dominant aerosol type indicates the dominance of dust over the latitudinal bands of 40°N-55°N and 30°N-35°N, and of carbonaceous aerosols near the surface between 43°N and 55°N, while primarily carbonaceous aerosols and secondarily sea salt aerosols are identified over the Mediterranean Sea (Fig. 7b), being in a satisfactory agreement with the CALIOP/CALIPSO observations. For the second selected CALIPSO track on 12Z of October $17^{th}$, dust and polluted dust are detected at 45°N-52°N and 30°N-35°N up to 3 km, with dust also observed in the upper troposphere (∼10 km) over France, while elevated smoke is identified over the North Sea at ∼4 km (Fig. 7c). The latitudinal bands of IFS dust dominance is consistent with CALIPSO, as well as the fire-originated aerosols (carbonaceous aerosols) which are found

over the North Sea above 2 km (Fig. 7d). Yet, over the Mediterranean carbonaceous aerosols prevail in IFS, while marine and dusty marine aerosols in CALIOP/CALIPSO.

## 3.3 Air quality

The implications of dust and biomass burning transport during the Ophelias's passage for European air quality are hereafter discussed. Figure 8 illustrates the CAMS global and regional $PM_{10}$ and $PM_{2.5}$ surface concentrations over Europe for the time period from 18Z on 15 October 2017 to 06Z on 17 October 2017. Overall, the spatial patterns of particulate matter concentrations in the CAMS products agree well, reproducing the plume of high aerosol loadings. Yet, there are some differences between the global and regional forecasts which are as follows: a) RegEns exhibits lower concentrations compared to IFS for both coarse and fine particulate matter, b) IFS has higher concentrations near the center of Ophelia probably due to higher sea salt contribution, c) the enhanced $PM_{10}$ and $PM_{2.5}$ concentrations over Portugal in IFS due to assimilated fire emissions (GFAS) are not seen in RegEns, d) RegEns exhibits higher concentrations in both $PM_{10}$ and $PM_{2.5}$ over the Black Sea and the Baltic States compared to IFS. The aforementioned inconsistencies are likely due to the different definition of $PM_{10}$ and $PM_{2.5}$ in IFS and each CAMS regional air quality model, the different aerosol schemes and horizontal resolution used in CAMS global and regional models, and the different assimilation approach in the CAMS regional air quality models relative to IFS.

To evaluate the CAMS forecast systems and the role of data assimilation in reproducing the enhancements in surface $PM_{10}$ and $PM_{2.5}$ concentrations, we employ ground-based measurements from the EMEP network from 8 rural stations (see Table 1 for further info and Fig. 9d for stations location) for intercomparison. For brevity, only the results from three EMEP stations are preseneted (Guipry FR0024R, France; Chilbolton Observatory GB1055R, Great Britain; Cabauw Wielsekade NL0644R, the Netherlands), while results for the rest of the stations are provided in the Supplement. Figure 9 presents the $PM_{10}$ surface concentrations from observations, IFS (day-1 to day-3 forecasts), IFS without data assimilation and RegEns (day-1 to day-3 forecasts), along with the percentage (%) contribution of each aerosol type to IFS $PM_{10}$ surface concentrations, during the time period form 10 October 2017 to 20 October 2017. At Guipry (France), after 12Z of October $14^{th}$ and up to October $16^{th}$ an increase in $PM_{10}$ surface concentrations is observed, also seen in IFS and RegEns; however the early peak in observations is not captured by CAMS forecast systems (Fig. 9a, top). The percentage contribution of each IFS aerosol type to IFS $PM_{10}$ concentrations indicates that the enhancement in $PM_{10}$ surface levels is initially (00Z on 15 October) due to organic matter aerosols ($\sim$75%), probably related to transported smoke from fires burning over the Iberian Peninsula (visual inspection of MODIS Fires and Thermal Anomalies product, source: NASA (2020)), while afterwards (21Z on 15 October) the transport of dust induces a significant contribution from dust aerosols ($\sim$53%) as well (Fig. 9a, bottom). On 12Z of October $16^{th}$ a sharp peak is depicted in observed $PM_{10}$ concentrations reproduced from both IFS and RegEns, yet overestimated from IFS, which seems to result from sea salt transport from the west and on the south of Ophelia (see flow at 850 hPa at Fig. 2i). IFS exhibits an increase of $PM_{10}$ levels on 12Z of October $18^{th}$ due to dust and organic matter aerosols which is not seen in RegEns and observations. Similar

performance for CAMS forecast systems is obtained for PM$_{2.5}$ over Guipry, except that the peak on 18 October in IFS is also confirmed from observations but to a lower extent (Fig. 10a). Further north and over Chilbolton Observatory (Great Britain), after 15Z of October 15$^{th}$ the observed PM$_{10}$ surface concentrations increase up to 38 $\mu$g/m$^3$, with IFS peaking with a 6 hours delay (Fig. 9b, top) mostly due to dust and organic matter aerosols (Fig. 9b, bottom). The secondary peak seen in observations on 12Z of October 16$^{th}$ is synchronously captured by RegEns and IFS, with IFS clearly overestimating it. A second wave of dust and biomass burning aerosols transport is affecting PM$_{10}$ surface concentrations over the site on 15Z of October 18$^{th}$ as revealed from IFS which is in agreement with observations but not seen in RegEns. Regarding PM$_{2.5}$, there is a similar level of agreement with that of PM$_{10}$ between the CAMS forecast systems and the observations (Fig. 10b). Finally, over Cabauw Wielsekade (the Netherlands), the observed PM$_{10}$ surface concentrations exhibit a more noisy structure, with CAMS forecast systems reproducing only the general pattern of variability, missing the observed peaks on 15Z of October 17$^{th}$ and 18$^{th}$ (Fig. 9c). Similarly for PM$_{2.5}$, where the observed peaks are more clear, IFS and RegEns capture the big picture of PM$_{2.5}$ increases (Fig. 10c). Results for the rest five EMEP stations are provided in Figs. S2 and S3 in the Supplement. Overall, RegEns exhibits a better performance in terms of temporal variability and bias compared to IFS. As depicted from the scatter plot in Fig. 11a the correlation coefficient of RegEns with observations is higher compared to that of IFS with observations (red points mostly lie above the dashed line of Fig. 11a), while the FGE values for RegEns are lower with respect to that of IFS (red points lie below the dashed line of Fig. 11b). Concerning the impact of data assimilation on IFS forecast of near surface particulate matter, as it is depicted from Fig. 11 there is no clear improvement neither in bias nor in the representation of the temporal variability. It is likely that the assimilation of total-column AOD does not have a strong impact on surface-layer concentrations, especially in regions with significant surface fluxes.

The capability of IFS and RegEns systems to forecast the observed PM$_{10}$ and PM$_{2.5}$ surface concentrations two and three days in advance, is finally discussed. As depicted in Figure 9, IFS day-2 and day-3 forecasts reproduce the distinct increases in observed PM$_{10}$ surface concentrations exhibiting similar FGE values but lower correlation scores (in most of the stations) compared to day-1 forecast (Fig. 11c and d). The same applies in the case of PM$_{2.5}$ (Fig. 10), except that the correlation scores for IFS day-2 and day-3 forecasts are not systematically lower than that of day-1 forecast (Fig. 11c and d). As regards the RegEns, although it fairly predicts the observed peaks in PM$_{10}$ and PM$_{2.5}$ up to three days in advance (Fig. 9 and 10), there is a systematic deterioration of its performance in terms of temporal variability over forecast time. More specifically, the correlation coefficient decreases from day-1 to day-2 forecast and from day-2 to day-3 forecast for almost all examined stations (Fig. 11e).

## 4   Conclusions

The main objectives of this work were to analyze a complex case study of aerosol transport over Europe driven by Storm Ophelia in mid-October 2017 in CAMS forecast systems, and assess their performance with respect to aerosol

atmospheric composition and air quality. To this end, CAMS forecast data were used along with satellite and ground-based observations. The most notable findings of this work can be summarized as follows:

- African dust aerosols were transported towards northwestern Europe on the east side of Storm Ophelia forming an extended dust plume reaching the Baltic States. On its passage, Ophelia lifted and drifted smoke aerosols from fires burning over the Iberian Peninsula carrying them over several European areas, such as France, Great Britain and the Netherlands. After 00Z of October $16^{th}$, dust/smoke aerosols are found on the warm/cold front of Ophelia being transported within its warm sector.

- The dependent evaluation against MODIS satellite observations reveals a satisfactory agreement of CAMS global $AOD_{550}$ (R=0.77 and FGE=0.4), while the comparison against the IFS no DA simulation indicates an improvement of IFS performance in forecasting $AOD_{550}$ (R increases from 0.72 to 0.77, and FGE decreases from 0.59 to 0.4), due to the application of data assimilation.

- The speciation of IFS $AOD_{550}$ fields indicates that African dust, organic matter from Iberian biomass burning and sea salt sprayed in the vicinity of Ophelia, are the major aerosol contributors to the plume of high $AOD_{550}$.

- The CALIOP/CALIPSO and the spatially and temporally collocated IFS cross sections of dominant aerosol types, reveal a good qualitative performance of IFS, as it generally manages to reproduce the type and location of aerosols during the passage of Ophelia.

- In terms of air quality, comparison with ground-based measurements from EMEP stations reveals that both IFS and RegEns are able to reproduce the observed signal of increase in $PM_{10}$ and $PM_{2.5}$ surface concentrations, yet exhibiting several inconsistencies on the quantitative and temporal representation of the observed peaks.

- For the examined event, the CAMS regional system seems to better predict the 3-h $PM_{10}$ and $PM_{2.5}$ surface concentrations compared to CAMS global system, with an increase/decrease of the correlation coefficient/FGE values in all examined stations for $PM_{10}$ (except Cabauw Wielsekade) and $PM_{2.5}$, respectively.

- As regards the role of data assimilation, even though it improves the CAMS global forecast performance for AOD there is no such indication for particulate matter at near surface.

- A deterioration of the RegEns forecast performance is found over forecast time for both $PM_{10}$ and $PM_{2.5}$, characterized by a decrease of the correlation coefficient for the vast majority of the examined stations, which is partially seen in IFS for the case of $PM_{10}$.

To summarize, the current analysis and evaluation study highlights that CAMS global forecast system is able to forecast the observed aerosol loadings over Europe induced from the transport of dust, biomass burning and sea salt aerosols through Ophelia's Storm passage in mid-October 2017. Forecasting coarse and fine particulate matter surface concentrations during Ophelia turns out to be more challenging, yet the CAMS global and regional systems exhibit a resonable performance, clearly open to further improvement and development in the direction of supporting social, economic and health policies.

*Author contributions.* DA designed the study, performed the analysis and wrote the paper. AKG provided the MODIS/Terra and Aqua satellite data, the CALIOP/CALIPSO satellite data and contributed in the analysis and writing of the paper. All authors contributed in reviewing and editing of the manuscript, and interpretation of the results. JF provided the IFS control simulation without the use of data assimilation. HE is coordinating CAMS84 (Global and regional a posteriori evaluation and quality assurance). EK is the Principal Investigator and coordinates the contribution of Aristotle University of Thessaloniki in CAMS84.

*Competing interests.* The authors declare that they have no conflict of interest.

*Acknowledgements.* This work is performed within the framework of the service element "CAMS84: Global and regional a posteriori evaluation and quality assurance" of the Copernicus Atmospheric Monitoring Services (CAMS). ECMWF is the operator of CAMS on behalf of the European Union. The CAMS84 work is financially supported by ECMWF via its main contractor, Royal Netherlands Meteorological Institute KNMI. NASA Goddard Space Flight Center (GSFC) Level-1 and Atmosphere Archive and Distribution System (LAADS) (LAADS, 2020) is acknowledged for making available the level-2 MODIS/Terra and Aqua Collection 6.1 aerosol datasets. In addition, the Atmospheric Science Data Center (ASDC) at NASA Langley Research Center (LaRC) is acknowledged for the data processing and distribution of the CALIPSO data used here (EARTHDATA, 2020). Also acknowledged is the European Monitoring and Evaluation Programme (EMEP) (EMEP, 2020) for distributing the near surface $PM_{10}$ and $PM_{2.5}$ concentration data. We acknowledge the use of imagery from the NASA Worldview application (NASA, 2020), part of the NASA Earth Observing System Data and Information System (EOSDIS). AUTH (Aristotle University of Thessaloniki) authors acknowledge the support of the scientific computing services of the AUTH-IT Center (AUTHIT, 2020). Finally, DA would like to acknowledge the free use of Python (Python, 2020) and Ferret (Ferret, 2020) software for the analysis and graphics of the paper.

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

**Table 1.** Observational data used in the present study.

| EMEP ground-based stations | | | | | |
|---|---|---|---|---|---|
| Name | Country | Code | Location | $PM_{10}$ | $PM_{2.5}$ |
| La Coulonche | France | FR0018R | 0.45° W,48.63° N | yes | no |
| Guipry | France | FR0024R | 1.84° W,47.83° N | yes | yes |
| Narberth | Great Britain | GB0043R | 4.69° W,51.78° N | yes | no |
| Auchencorth Moss | Great Britain | GB0048R | 3.24° W,55.79° N | yes | no |
| Chilbolton Observatory | Great Britain | GB1055R | 1.44° W,51.15° N | yes | yes |
| Kollumerwaard | the Netherlands | NL0009R | 6.28° E,53.33° N | yes | yes |
| De Zilk | the Netherlands | NL0091R | 4.5° E,52.3° N | yes | yes |
| Cabauw Wielsekade | the Netherlands | NL0644R | 4.92° E,51.97° N | yes | yes |

| Satellite observations | | | | | |
|---|---|---|---|---|---|
| Sensor/Satellite | Type | Overpass | Coverage | Resolution | Product |
| MODIS/Terra | passive | 10:30 LT | daily | 10 x 10 km$^2$ | $AOD_{550}$ |
| MODIS/Aqua | passive | 13:30 LT | daily | 10 x 10 km$^2$ | $AOD_{550}$ |
| CALIOP/CALIPSO | active | $\sim$ 10:30 LT | 16-day repeat cycle | hor.: 5 km, ver.: 60 m (<20 km) | aerosol subtype |

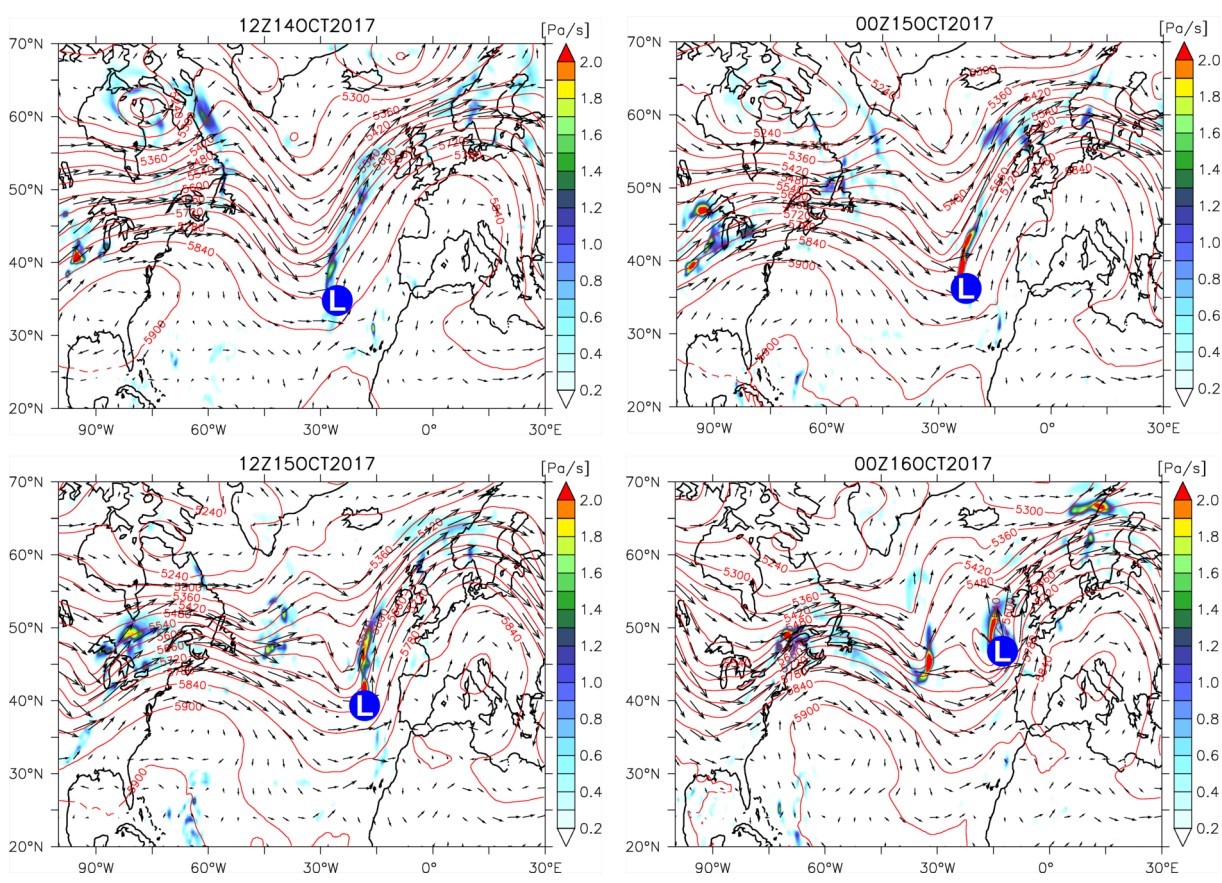

**Figure 1.** IFS geopotential height (in gpm; red contours), negative of omega vertical velocity (-dp/dt) (in Pa/s, color shaded) and wind direction (black vectors) at 500 hPa, during the period from 12:00Z on 14 October 2017 to 00:00Z on 16 October 2017 (12 h interval) (a, b, c, d). Also shown is the location of the surface low pressure system denoted with the capital letter L.

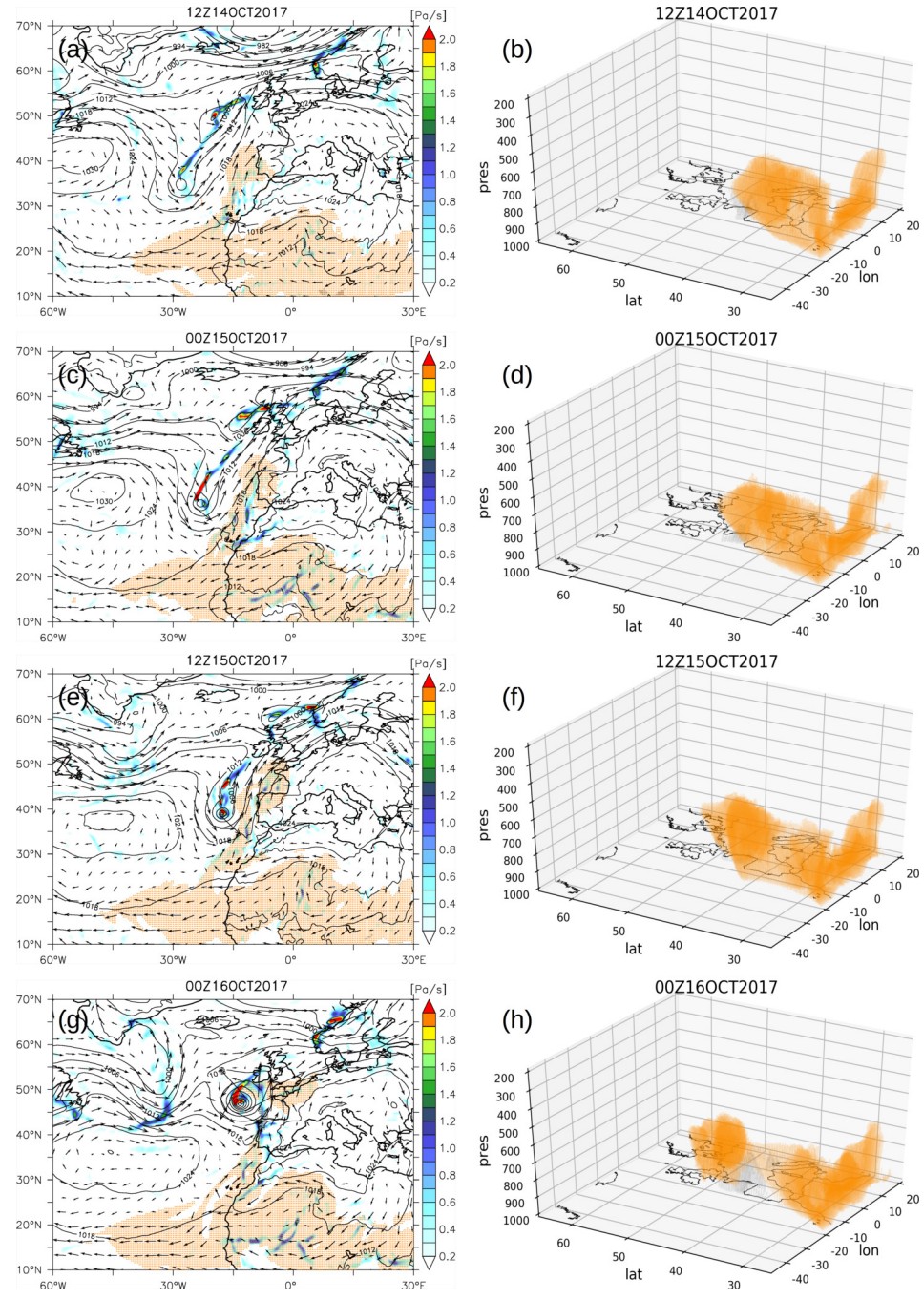

**Figure 2.** First part of the figure

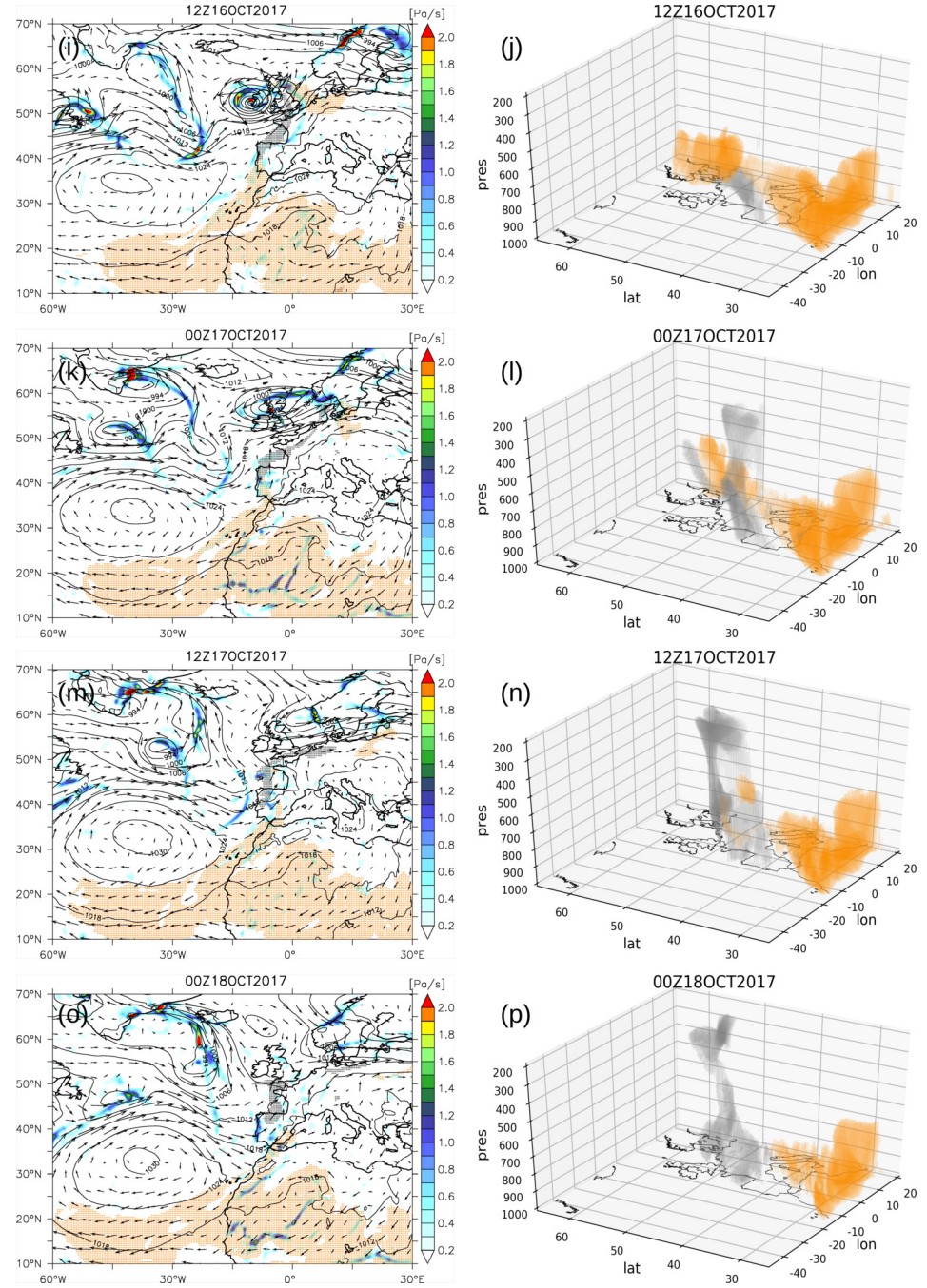

**Figure 2 (Continued).** IFS mslp (in hPa; black contours), negative of omega vertical velocity (-dp/dt) (in Pa/s, color shaded), wind direction (black vectors), dust with mass mixing ratio exceeding 80 $\mu$g/kg (orange shaded) and CO with volume mixing ratio exceeding 200 ppbv (grey shaded) at 850 hPa during the period from 12:00Z on 14 October 2017 to 00:00Z on 18 October 2017 (12 h interval) (Figures 2a, 2c, 2e, 2g, 2i, 2k, 2m, 2o). Three-dimensional (longitude, latitude, pressure (hPa)) spatial distribution of IFS dust with mass mixing ratio exceeding 80 $\mu$g/kg (orange shaded) and IFS carbon monoxide (CO) with volume mixing ratio exceeding 200 ppbv (grey shaded) during the same period (Figures 2b, 2d, 2f, 2h, 2j, 2l, 2n, 2p).

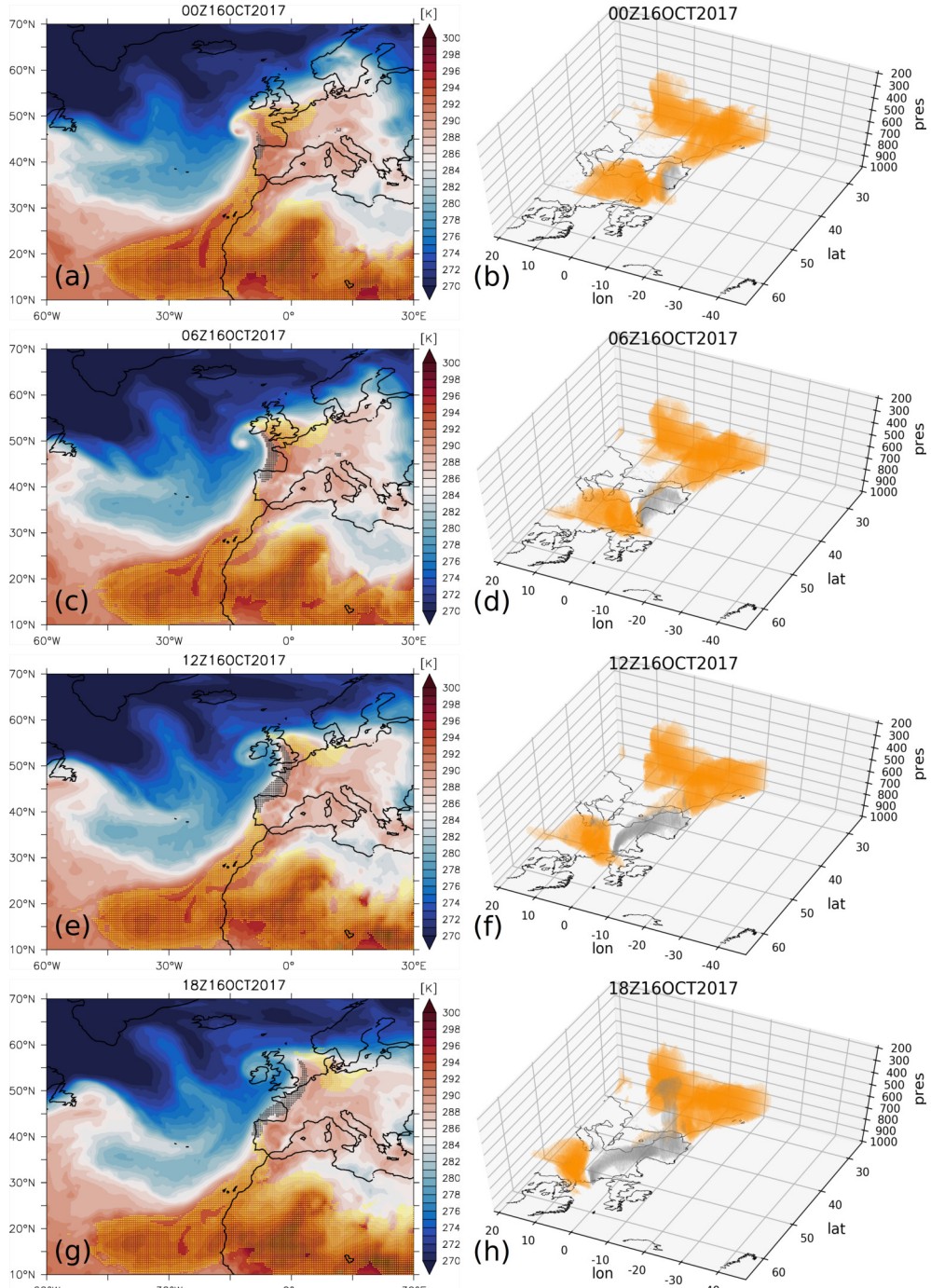

**Figure 3.** IFS temperature (in K; color shaded), dust with mass mixing ratio exceeding 80 $\mu$g/kg (orange shaded) and CO with volume mixing ratio exceeding 200 ppbv (grey shaded) at 850 hPa during the period from 00:00Z on 16 October 2017 to 18:00Z on 16 October 2017 (12 h interval) (Figures 3a, 3c, 3e, 3g). Three-dimensional (longitude, latitude, pressure (hPa)) spatial distribution of IFS dust with mass mixing ratio exceeding 80 $\mu$g/kg (orange shaded) and IFS CO with volume mixing ratio exceeding 200 ppbv (grey shaded) during the same period (Figures 3b, 3d, 3f, 3h).

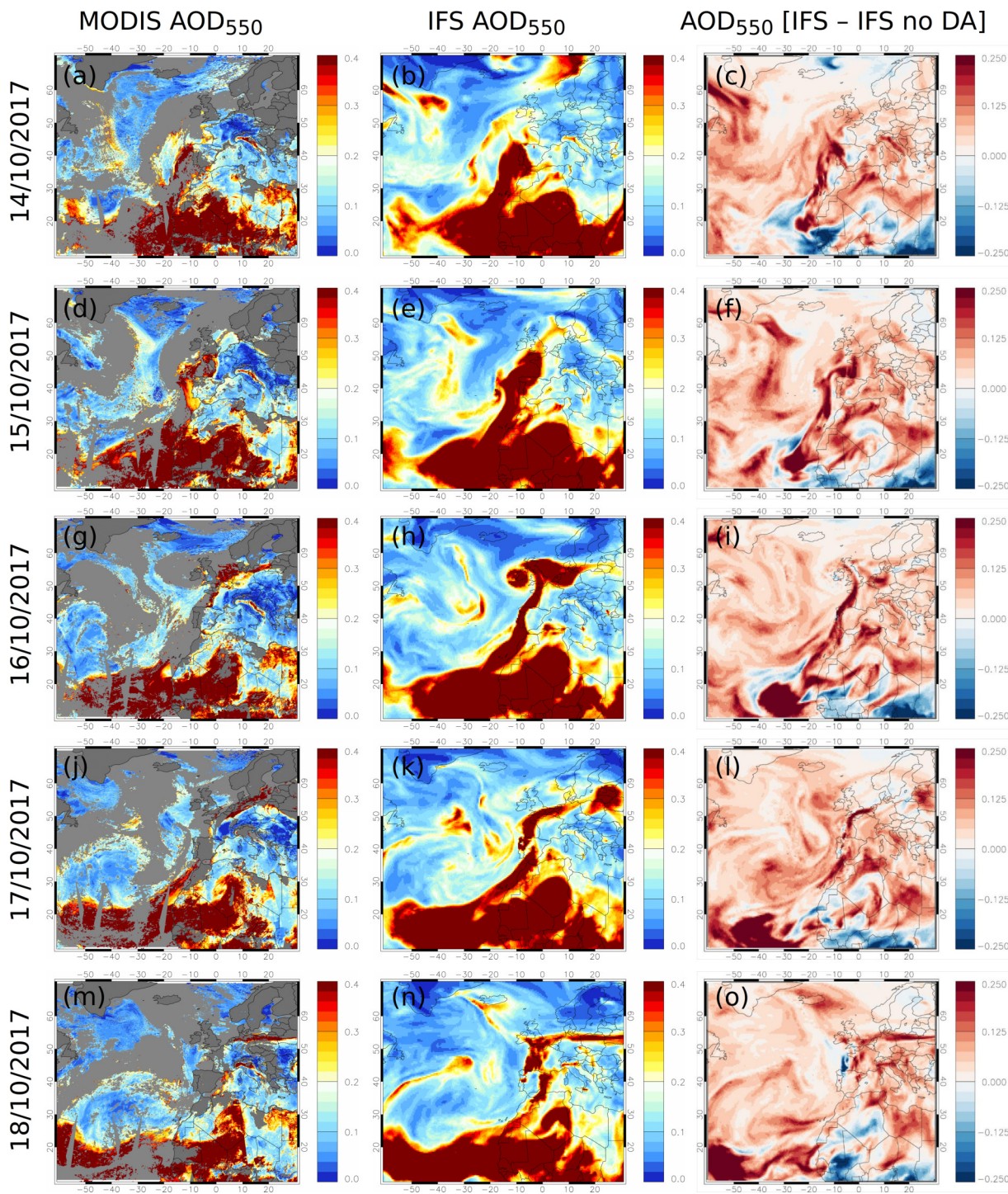

**Figure 4.** MODIS/Terra and Aqua Collection 6.1 AOD$_{550}$ during the period from 14 October 2017 to 18 October 2017 (Figures 4a, 4d, 4g, 4j, 4m). IFS AOD$_{550}$ (Figures 4b, 4e, 4h, 4k, 4n) and differences between IFS and IFS no DA AOD$_{550}$ (Figures 4c, 4f, 4i, 4l, 4o) at 12:00Z of the respective day.

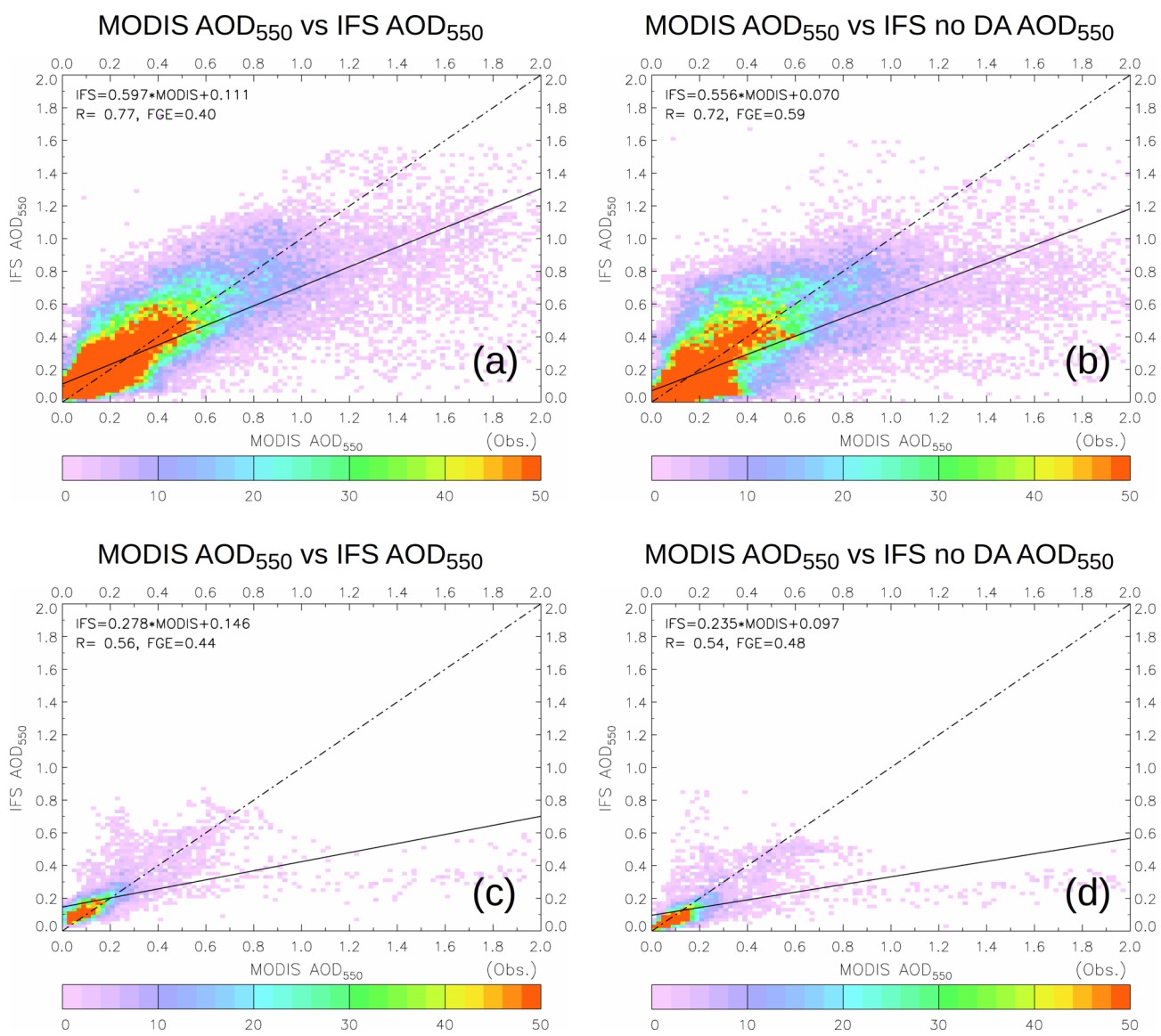

**Figure 5.** Comparison of spatially and temporally collocated IFS and MODIS/Terra and Aqua Collection 6.1 $AOD_{550}$ data for: the whole domain of Fig. 4, and IFS including data assimilation (a), IFS without data assimilation (b); and the box region appearing in Fig. 9d, and IFS including data assimilation (c), IFS without data assimilation (d). The examined dates are those appearing in Fig. 4. The color scale corresponds to the number of IFS-MODIS collocation points that fall within 0.02 x 0.02 $AOD_{550}$ bins. The solid line is the regression line of the IFS-MODIS data and the dashed-dotted line is the 1:1 line. The slope and the intercept of the regression line, the correlation coefficient (R) and the fractional gross error (FGE) are also shown.

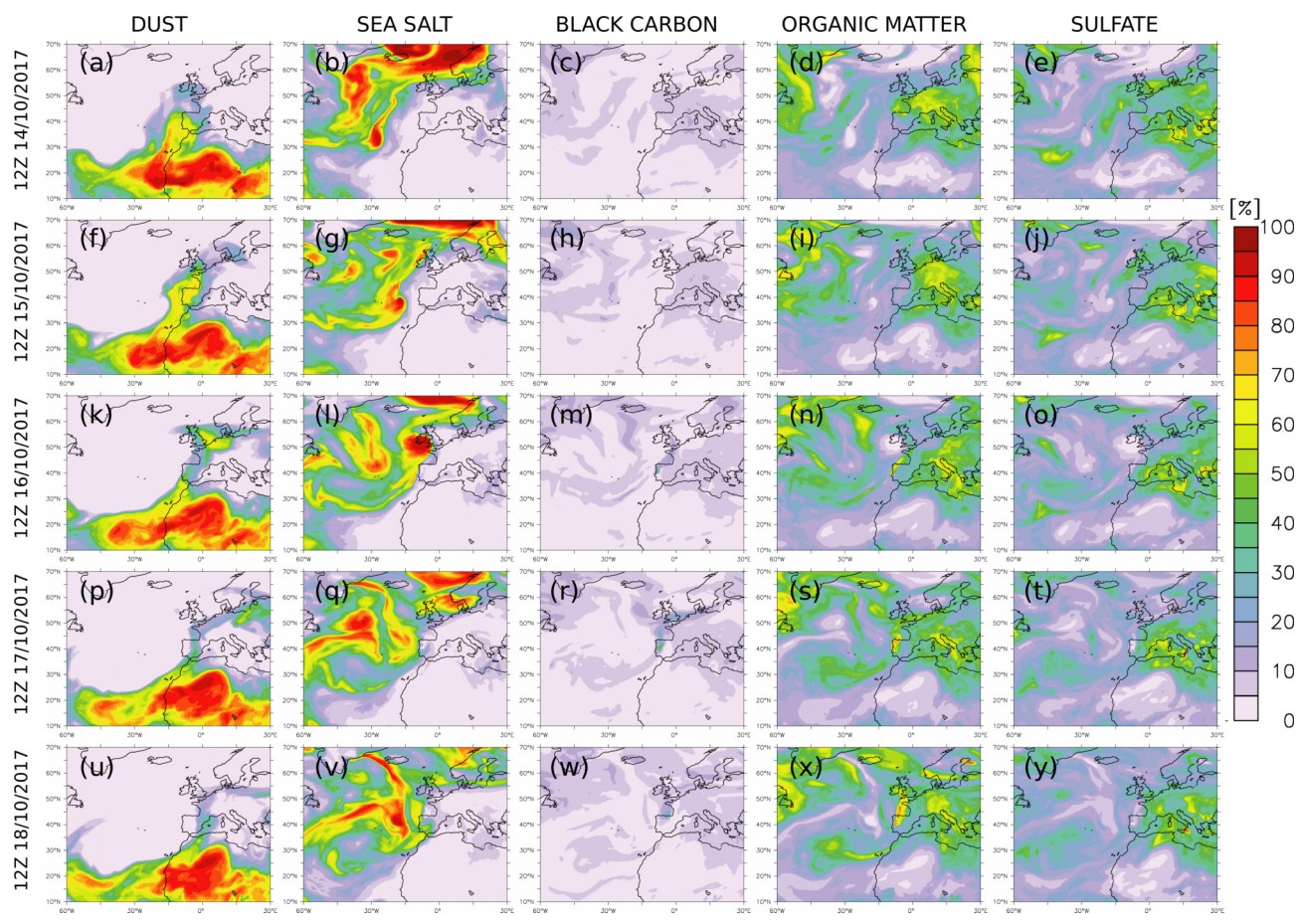

**Figure 6.** Percentage contribution of dust (Figures 6a, 6f, 6k, 6p, 6u), sea salt (Figures 6b, 6g, 6l, 6q, 6v) , black carbon (Figures 6c, 6h, 6m, 6r, 6w), organic matter (Figures 6d, 6i, 6n, 6s, 6x) and sulfate (Figures 6e, 6j, 6o, 6t, 6y) aerosols to the IFS total $AOD_{550}$ values appearing in Fig. 4.

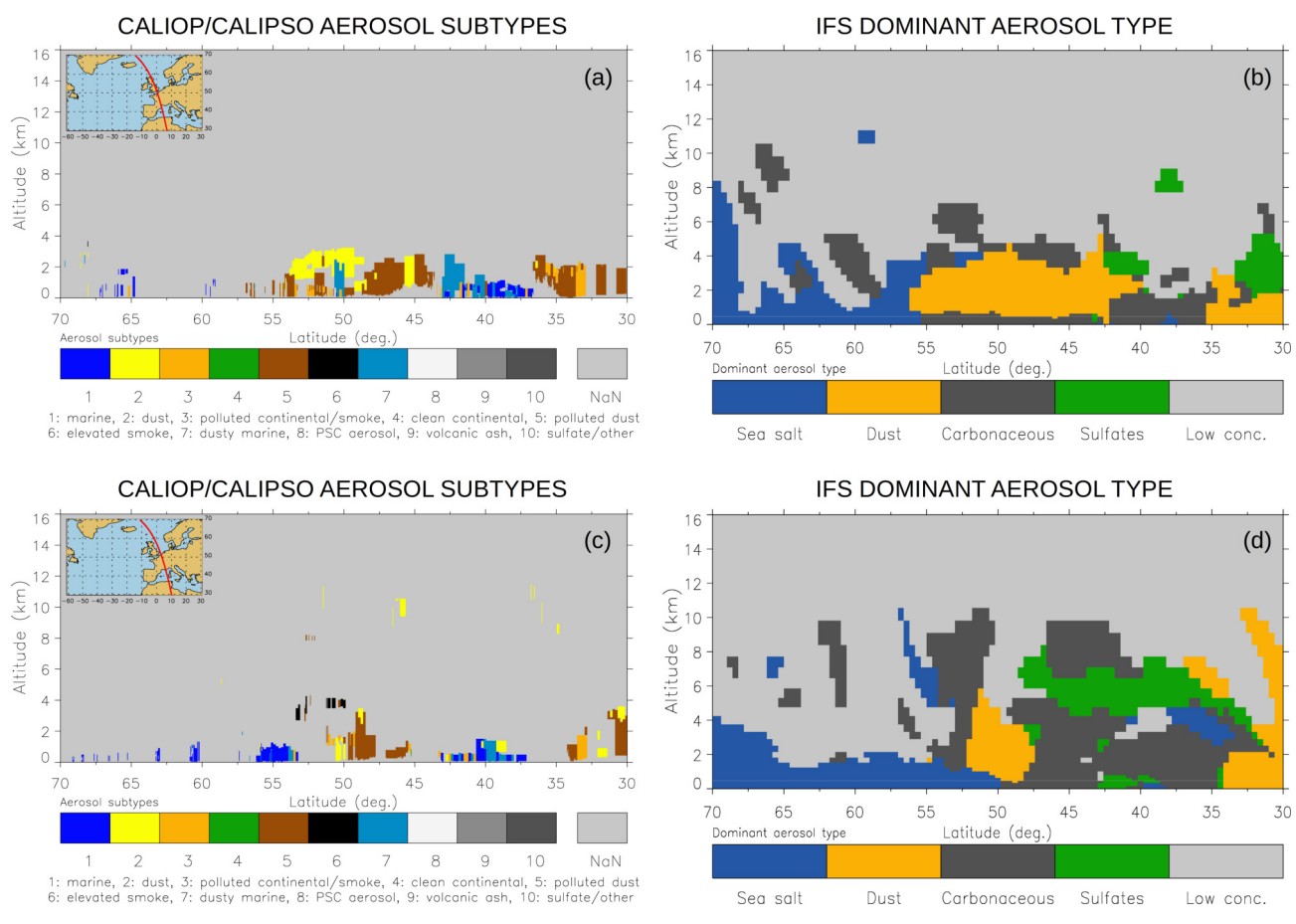

**Figure 7.** Vertical cross-sections of CALIOP/CALIPSO aerosol subtypes and IFS dominant aerosol type on 12Z 15 October 2017 (a, b) and on 12Z 17 October 2017 (c, d). The embedded maps in Figs. 7a and c depict the respective CALIOP/CALIPSO tracks.

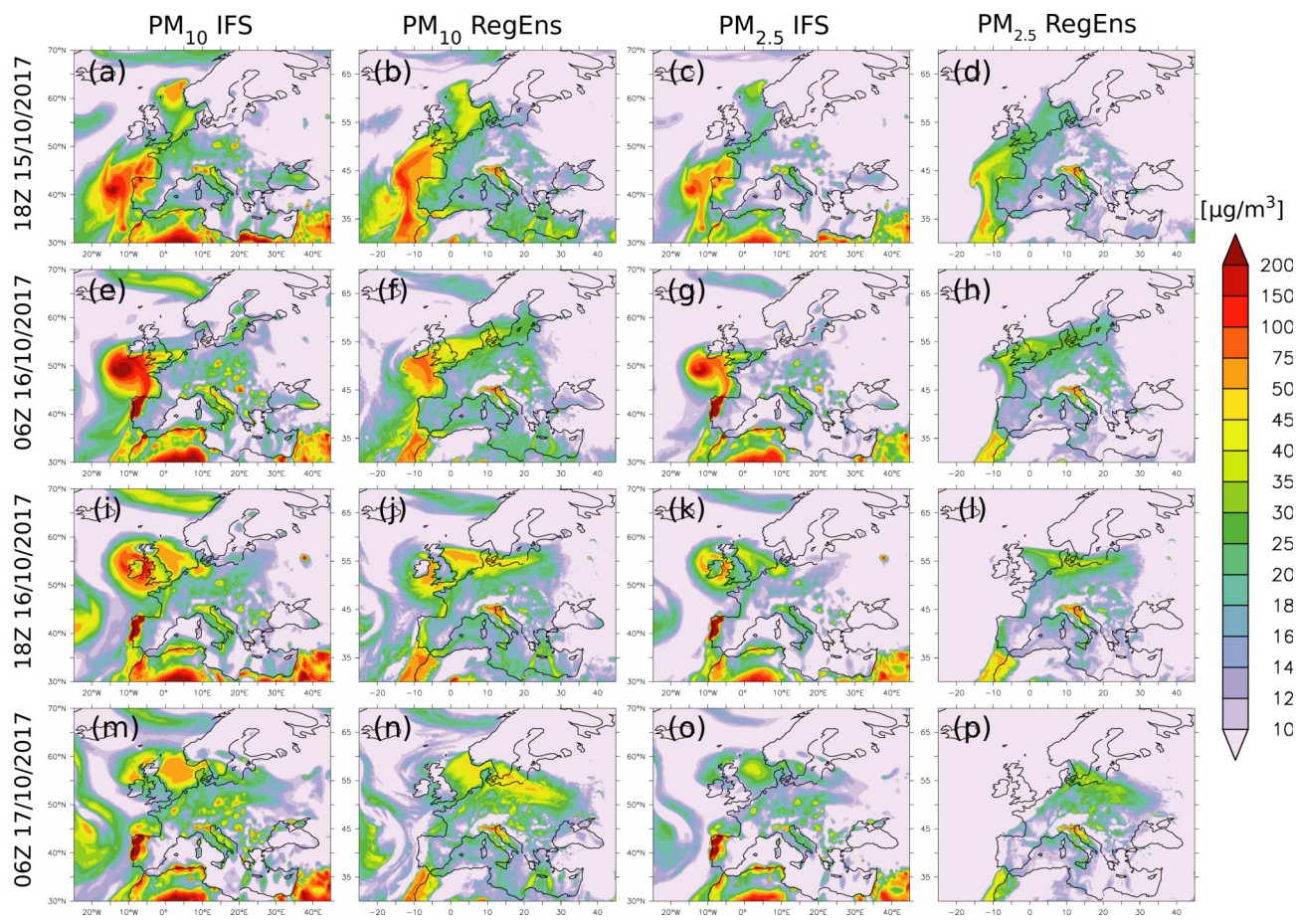

**Figure 8.** $PM_{10}$ surface concentrations ($\mu g/m^3$) for IFS (Figures 8a, 8e, 8i, 8m) and RegEns (Figures 8b, 8f, 8j, 8n) for the period from 18:00Z on 15 October 2017 to 06:00Z on 17 October 2017. $PM_{2.5}$ surface concentrations ($\mu g/m^3$) for IFS (Figures 8c, 8g, 8k, 8o) and RegEns (Figures 8d, 8h, 8l, 8p) for the same period.

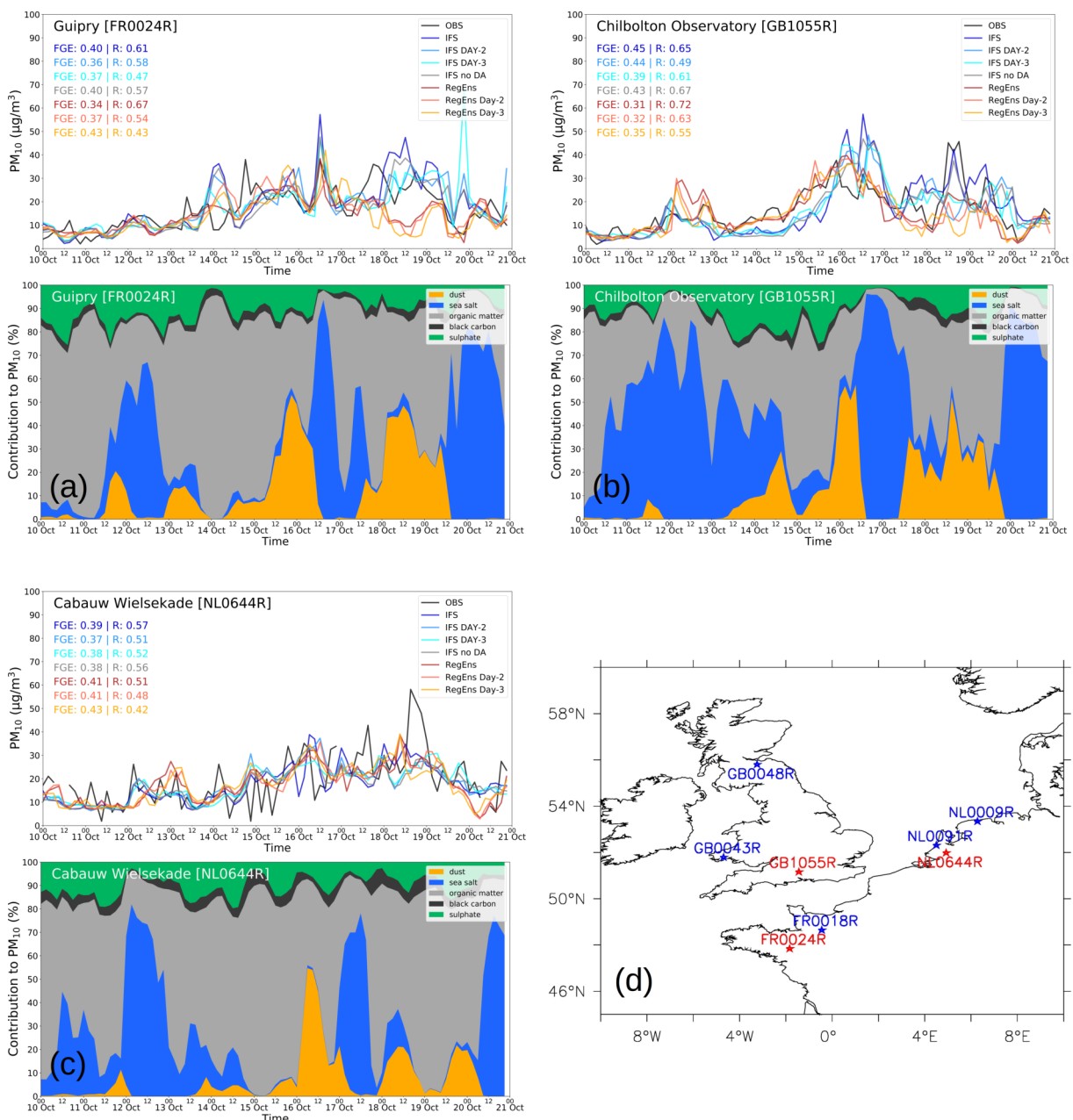

**Figure 9.** Observed (black), IFS (blue), IFS day-2 (light blue), IFS day-3 (aqua), IFS no DA (grey), RegEns (red), RegEns day-2 (light red)and RegEns day-3 (orange) 3-hourly timeseries of PM$_{10}$ surface concentrations ($\mu$g/m$^3$) (top), and percentage (%) contribution of aerosol type to IFS PM$_{10}$ surface concentrations (bottom) for Guipry (FR0024R), France (a); Chilbolton Observatory (GB1055R), Great Britain (b); Cabauw Wielsekade (NL0644R), the Netherlands (c). The locations of the presented EMEP stations in the manuscript (red) and the Supplement (blue) are also shown (d).

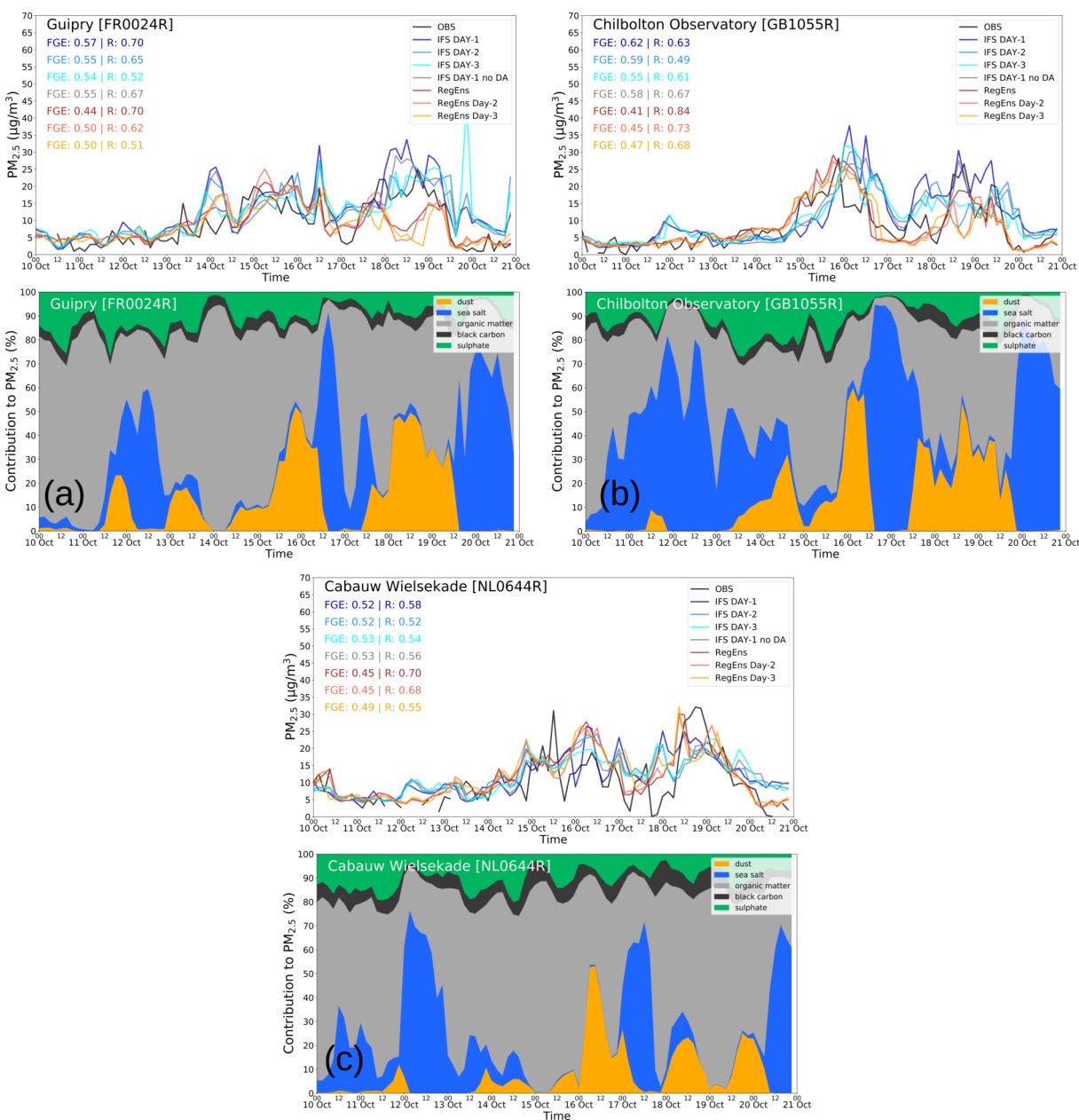

**Figure 10.** Observed (black), IFS (blue), IFS day-2 (light blue), IFS day-3 (aqua), IFS no DA (grey), RegEns (red), RegEns day-2 (light red)and RegEns day-3 (orange) 3-hourly timeseries of of $PM_{2.5}$ surface concentrations ($\mu g/m^3$) (top), and percentage (%) contribution of aerosol type to IFS $PM_{2.5}$ surface concentrations (bottom) for Guipry (FR0024R), France (a); Chilbolton Observatory (GB1055R), Great Britain (b); Cabauw Wielsekade (NL0644R), the Netherlands (c).

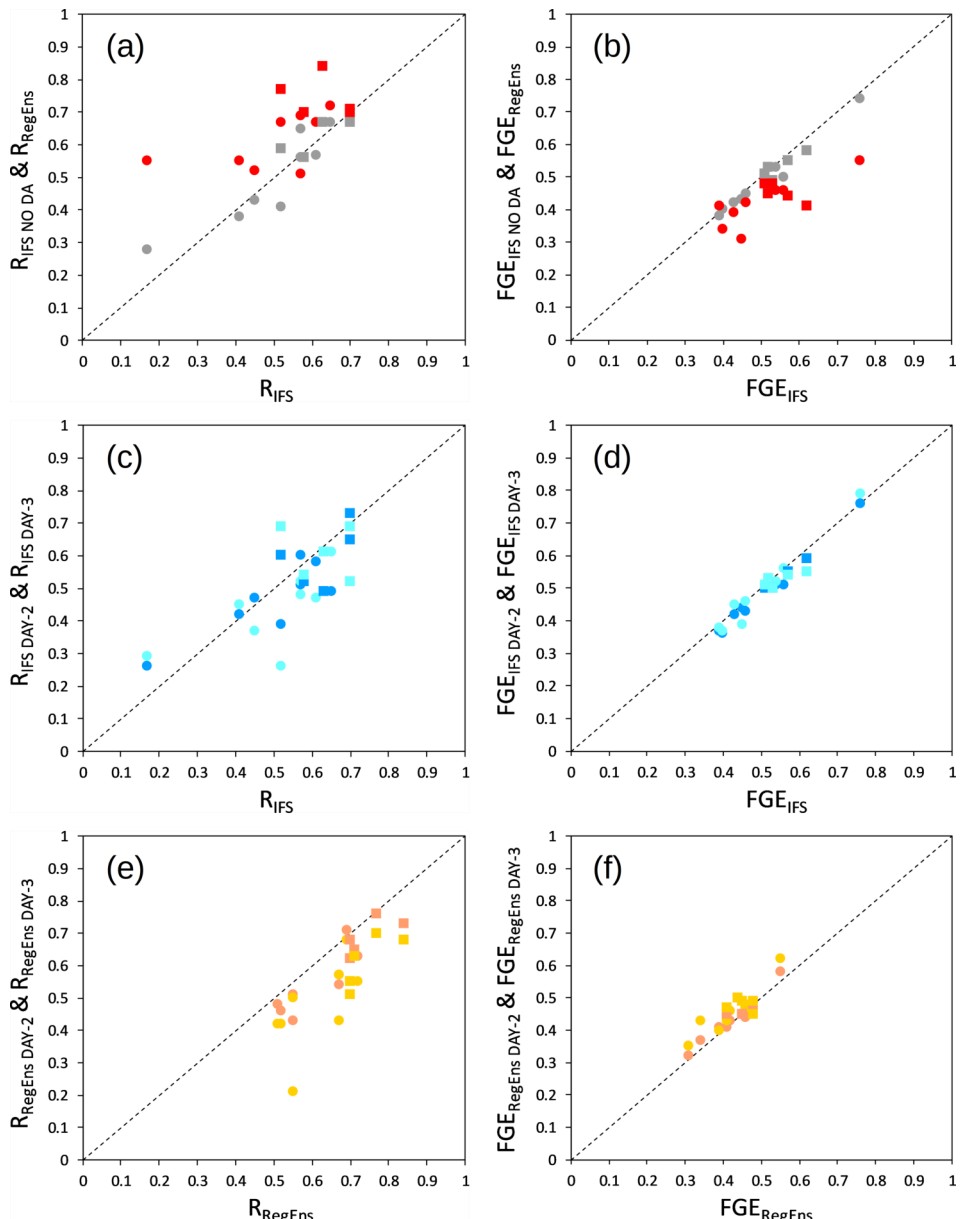

**Figure 11.** (a) Comparison between the correlation coefficients IFS vs OBS ($R_{IFS}$) and IFS no DA vs OBS ($R_{IFS\,NO\,DA}$) (grey symbols), as well as between IFS vs OBS ($R_{IFS}$) and RegEns vs OBS ($R_{RegEns}$) (red symbols) for PM$_{10}$ (circles) and PM$_{2.5}$ (squares). (b) Same as (a) but for FGE. (c) Comparison between the correlation coefficients IFS vs OBS ($R_{IFS}$) and IFS DAY-2 vs OBS ($R_{IFS\,DAY-2}$) (light blue symbols), as well as between IFS vs OBS ($R_{IFS}$) and IFS DAY-3 vs OBS ($R_{IFS\,DAY-3}$) (aqua symbols) for PM$_{10}$ (circles) and PM$_{2.5}$ (squares). (d) Same as (c) but for FGE. (e) Comparison between the correlation coefficients RegEns vs OBS ($R_{RegEns}$) and RegEns DAY-2 vs OBS ($R_{RegEns\,DAY-2}$) (light red symbols), as well as between RegEns vs OBS ($R_{RegEns}$) and RegEns DAY-3 vs OBS ($R_{RegEns\,DAY-3}$) (orange symbols) for PM$_{10}$ (circles) and PM$_{2.5}$ (squares). (f) Same as (e) but for FGE. The results for PM$_{10}$ (PM$_{2.5}$) are obtained from the 8 (5) EMEP stations described in Table 1.