# Peer review of "A complex aerosol transport event over Europe during the 2017 Storm Ophelia in CAMS forecast systems: analysis and evaluation"

_Atmospheric Chemistry and Physics, 2020_

## Referee Comment (RC1) · Anonymous Referee #1 · 7 Jul 2020

This is a very well, and concisely written evaluation of the ability of the CAMS global and regional systems for forecasting a rather particular aerosol event, being a combination of high loadings of dust, biomass burning and sea salt particles. I find interesting to learn that the IFS forecasts excluding data-assimilation already show similar performance as the configuration including data-assimilation. Does this either point at a very good forecast model strength, or rather at deficiencies in the data assimilation setup?

I find it only unfortunate that authors only assess the first-day forecasts. This only a limited view of the forecast capability of the system is presented. It would have been very interesting if the authors would have shown the (likely decay in) forecast capability

for the second and third day forecasts, as this would give a better handle for 'citizens and policy-makes' on the reliability of forecasts on a time scale where they are able to take action, which is one of the key objectives of this study. Is it possible for authors to make statements on this?

Apart from this, I only have some small comments.

Page 4, l 10L 'day-1 forecasts'. Are these the forecasts initiated at 0h00? Just because the CAMS operational system provides currently two forecasts per day.

Page 4. The authors provide empirical formulae to compute the PM10 and PM2.5 concentrations from the global system, which are crucial for the definition of the contribution of aerosol to the air quality statistics. These equations appear rather empirical, and also different depending on the model version. Can the authors give some more background information as to how these metrics are designed? Also, to what extent does the definition of this metric contribute to differences seen in Figure 8?

Page 5, l3: "CAMS regional models assimilate PM10 and PM2.5" : does this hold for all CAMS regional models, or only for some? Also, are PM10 and PM2.5 the modeled tracer fields in the regional models, or are they computed from underlying aerosol composition fields, as is the case for the global model? In summary, could the authors elaborate a little more on how PM is modeled in the regional system?

page 7, line 26: "reducing the bias"-> reducing the error

page 8, line 1: "the percentage" -> the modeled percentage

page 10, line 30: "implementation": I would rather write "application".

page 11, line 9: "The CAMS regional system seems to better predict the. . ..": why not write something like: "for this event the CAMS regional system shows better .."
* * *

---

## Referee Comment (RC2) · Anonymous Referee #3 · 13 Jul 2020

The authors analyse a remarkable event of combined desert dust and biomass burning aerosol transport over Europe using observations and model results, and in the process evaluate some aspects of the CAMS air quality forecast system. Studying such events is important because of their quite drastic impact on the European air quality, potential implications on clouds and weather, and impairment of solar energy yield. Their distinct air quality signatures are good benchmarks for forecast systems. The subject is relevant and within the scope of ACP (there is some overlap with the GMD scope regarding the model evaluation aspect), the article is well organised and written so that I recommend publication after addressing the following comments:

[Figure]

Page 1, line 11: Throughout the manuscript you use the correlation (coefficient) r, here you present the shared variance (and I assume you are referring to r^2). Because the term is less common or even used with different meaning, and to be consistent with the main text, I suggest to stick to r.

Page 1, line 13: Please expand IFS here.

Section 2.1: Some more information about the aerosol representation in the model would be helpful: does the model assume the different aerosol types to be externally mixed, both regarding the aerosol optical properties and regarding any chemical or physical interactions between different components? This would be interesting to know especially since the event under consideration involves dust, biomass burning aerosol and sea salt.

Page 4, line 8 and 9: The units should be micro metres not metres and it has to be mentioned that the numbers represent radii not diameters. What bins are used for the other aerosol types, e.g., what is BC1 vs. BC2?

Page 4, Eqs. (1) and (2): I understand these equations follow recommendations elsewhere, but could you indicate in the text where the factors come from? Also, SS3 is not considered for PM2.5 (and PM10), because the radius is > 5 um, but according to the intervals provided above, not considering DD3 for PM2.5 seems to ignore dust particles between 0.9 and 1.25 um radius.

Page 5, line 10: Please expand LT once

Page 5, lines 28ff: Which selection criteria for the stations where used exactly?

Page 7, line 26: The FGE deserves to be introduced by an equation, moreover uppercase is more common.

Page 7, lines 30 to 34: How can this conclusion be aligned with Fig. 4, where in many regions with low AOD the AOD is enhance by data assimilation?

Figs. 9 and 10: I find the similarity of the PM10 and PM2.5 composition surprising, normally I would expect a higher dust and sea salt fraction in PM10 than in PM2.5. Does this indicate some limitation of the model? After all, from Eqs. (1) and (2) it is clear that the contributions in Fig. 9 and 10 cannot be that different, and the largest particles (SS3 and DD3) are not relevant at all, but is that realistic? Do the stations provide the PM2.5 and/or PM10 composition for comparison, or is there any other suitable data source to validate this aspect?

Generally, I believe links in footnotes and within the text are supposed to be be moved to the References section to comply with the journal standards.

---

## Referee Comment (RC3) · Anonymous Referee #2 · 16 Jul 2020

The authors analyse an interesting meteorological/aerosol event named Ophelia, that took place in mid-October 2017 over western Europe. They use CAMS global day-1 forecast data to assess the aerosol spatiotemporal distribution and determine the source and type of aerosol species. Validating the performance of CAMS in these kind of events is quite crucial to understand the current limitations in CAMS aerosol forecast system and the possible upgrades that need to be planned (e.g. assimilating more and different kinds of aerosol observations). The manuscript is well written providing all the necessary information to introduce the event to the reader. I would recommend for publication in ACP after addressing some minor comments below:

[Figure]

CAMS AOD is evaluated using dependent observations (as mentioned at P7, L10-11) from MODIS Dark Target and Deep Blue algorithms showing reasonable agreement with the observations and improvement in both correlation and bias in comparison to the no assimilation experiment. The paper does not provide a comparison with independent observations to prove that data assimilation improves AOD. I would suggest to mention again in conclusions that this is a dependent evaluation (P10, L28-31) or perform a small analysis using independent observations (e.g. AERONET AOD) to prove that CAMS AOD really improves, although the latter option may be out of the scope of this paper.

Figure 6 and discussion starting at P8, L1: A very interesting analysis highlighting the contribution of each species to AOD. Nevertheless the contribution of each species to AOD may differ depending on the aerosol-species optical properties. Black carbon AOD is low (and Absorption AOD high) in comparison to the other species. A 15% to 25% contribution of black carbon AOD to the total AOD might seem mediocre, but climatologically is only observed during fire season in the Tropical band. Maybe the authors would like to comment on that.

Figure 7 and discussion starting at P8, L17: Although this is more of a qualitative comparison to discuss the vertical distribution of aerosols per species, the depiction of CALIPSO and IFS are very different above 4km in both tracks. Is it possible to conclude something about the observations or the model? For example is CALIPSO unable to retrieve trustworthy measurements during the pass of Ophelia above 4km or is IFS overestimating aerosol mixing ratio above a certain height?

---

## Author Comment (AC1) · 18 Sep 2020

We would like to thank Reviewer #1 for her/his time devoted and the constructive and helpful comments to which we will respond point by point.

This is a very well, and concisely written evaluation of the ability of the CAMS global and regional systems for forecasting a rather particular aerosol event, being a combination of high loadings of dust, biomass burning and sea salt particles. I find interesting to learn that the IFS forecasts excluding data-assimilation already show similar performance as the configuration including data-assimilation. Does this either point at a very good forecast model strength, or rather at deficiencies in the data assimilation setup?

We thank the Reviewer for the general comment. In fact, the use of data assimilation improves the performance of CAMS system in AOD forecast, yet it seems to leave relatively unaffected the $PM_{10}$ and $PM_{2.5}$ forecast near the surface. Both speculations seem valid. On the one hand, IFS is a state-of-the-art model that apparently reproduces the aerosol transport and the induced increase in particulate matter (PM) surface concentrations, without the use of data assimilation. On the other hand, it seems that the data assimilation of a columnar aerosol product (AOD) has no substantial effect on surface PM air quality forecast, for the examined event.

I find it only unfortunate that authors only assess the first-day forecasts. This only a limited view of the forecast capability of the system is presented. It would have been very interesting if the authors would have shown the (likely decay in) forecast capability for the second and third day forecasts, as this would give a better handle for 'citizens and policy-makes' on the reliability of forecasts on a time scale where they are able to take action, which is one of the key objectives of this study. Is it possible for authors to make statements on this?
Apart from this, I only have some small comments.

We agree with the suggestion raised by the Reviewer and thus in the Revised Manuscript (RM) we have included IFS and RegEns day-2 and day-3 forecasts of $PM_{10}$ and $PM_{2.5}$ in Figures 9, 10, S2 and S3. Additionally, we have extended Figure 11 including the results from day-2 and day-3 forecasts. Accordingly, we have modified/added several parts in the RM.

The following paragraph is included in the RM at P11, L21-29:

**"The capability of IFS and RegEns systems to forecast the observed $PM_{10}$ and $PM_{2.5}$ surface concentrations two and three days in advance, is finally discussed. As depicted in Figure 9, IFS day-2 and day-3 forecasts reproduce the distinct increases in observed $PM_{10}$ surface concentrations exhibiting similar FGE values but lower correlation scores (in most of the stations) compared to day-1 forecast (Fig. 11c and d). The same applies in the case of $PM_{2.5}$ (Fig. 10), except that the correlation scores for IFS day-2 and day-3 forecasts are not systematically lower than that of day-1 forecast (Fig. 11c and d). As regards the RegEns, although it fairly predicts the observed peaks in $PM_{10}$ and $PM_{2.5}$ up to three days in advance (Fig. 9 and 10), there is a systematic deterioration of its performance in terms of temporal variability over forecast**

**time. More specifically, the correlation coefficient decreases from day-1 to day-2 forecast and from day-2 to day-3 forecast for almost all examined stations (Fig. 11e)."**

The following sentence is included in the Conclusions of the RM at P12, L25-27:

**"A deterioration of the RegEns forecast performance is found over forecast time for both PM$_{10}$ and PM$_{2.5}$, characterized by a decrease of the correlation coefficient for the vast majority of the examined stations, which is partially seen in IFS for the case of PM$_{10}$."**

We have modified a part of the Abstract, see P1, L15-17 at the RM:

**"Regarding the footprint on air quality, both CAMS global and regional forecast systems are generally able to reproduce the observed signal of increase in surface particulate matter concentrations. The regional component performs better in terms of bias and temporal variability, with the correlation deteriorating over forecast time."**

Page 4, l 10L 'day-1 forecasts'. Are these the forecasts initiated at 0h00? Just because the CAMS operational system provides currently two forecasts per day.

Yes. Both CAMS global and regional forecasts used in the paper are initiated at 00:00 UTC. In the RM, we have included the respective information at P4, L8 and P5, L12 as follows: "**(initiated at 00:00Z)**".

Page 4. The authors provide empirical formulae to compute the PM10 and PM2.5 concentrations from the global system, which are crucial for the definition of the contribution of aerosol to the air quality statistics. These equations appear rather empirical, and also different depending on the model version. Can the authors give some more background information as to how these metrics are designed? Also, to what extent does the definition of this metric contribute to differences seen in Figure 8?

These are indeed empirical formulae which are updated when necessary with IFS cycle updates. Regarding the PM$_{10}$ and PM$_{2.5}$ formulae, the factors applied in sea salt bins are used to transform sea salt from 80% relative humidity ambient conditions to dry, while the rest correspond to the fraction of each aerosol type included in PM$_{10}$ and PM$_{2.5}$. More information on the description and evaluation of the aerosol scheme used in IFS is provided by Remy et al. (2019) which we have included in the RM at P4, L18-19 as follows: **"A detailed description and evaluation of the aerosol scheme used in IFS can be found in Remy et al. (2019)."** The differences seen in Figure 8 are likely also due to the definition of PM$_{10}$ and PM$_{2.5}$ in IFS and CAMS regional models, which is already stated in the manuscript as **"The aforementioned inconsistencies are likely due to the different definition of PM$_{10}$ and PM$_{2.5}$ in IFS and each CAMS regional air quality model,.."** (see P10, L12-13 of the RM). A quantitative estimation of this is beyond the scope of this paper, yet that would be an interesting task for a future study.

Page 5, l3: "CAMS regional models assimilate PM10 and PM2.5" : does this hold for all CAMS regional models, or only for some? Also, are PM10 and PM2.5 the modeled tracer fields in the regional models, or are they computed from underlying aerosol composition fields, as is the case

for the global model? In summary, could the authors elaborate a little more on how PM is modeled in the regional system?

Indeed, not all CAMS regional models were assimilating $PM_{10}$ and $PM_{2.5}$ at the time of the event. More specifically, CHIMERE and EURAD were assimilating both $PM_{10}$ and $PM_{2.5}$, MOCAGE only $PM_{10}$, while SILAM and MATCH were assimilating only $PM_{2.5}$. $PM_{10}$ and $PM_{2.5}$ in the regional models are calculated using underlying aerosol species, as it is the case with the CAMS global model. However, each model is free to choose how exactly this is done, as the aerosol schemes and thus the aerosol species of each model differ. These details have been added in the RM at P5, L3-8: **"Several CAMS regional models assimilate $PM_{10}$ and $PM_{2.5}$ surface observations from various stations of the EEA's (European Environment Agency) Air Quality e-reporting database, but not satellite aerosol products. More specifically, during the period of interest (October 2017), CHIMERE and EURAD were assimilating both $PM_{10}$ and $PM_{2.5}$, MOCAGE only $PM_{10}$ and, finally, SILAM and MATCH were assimilating only $PM_{2.5}$. $PM_{10}$ and $PM_{2.5}$ concentrations in the regional models are calculated using simulated aerosol fields specific to each regional model."**

page 7, line 26: "reducing the bias"-> reducing the error
Done.

page 8, line 1: "the percentage" -> the modeled percentage
Done.

page 10, line 30: "implementation": I would rather write "application".
Done.

page 11, line 9: "The CAMS regional system seems to better predict the: : :." : why not write something like: "for this event the CAMS regional system shows better .."
We have included the phrase **"For the examined event,"** in the beginning of the respective sentence (P12, L20).

**References**

Rémy, S., Kipling, Z., Flemming, J., Boucher, O., Nabat, P., Michou, M., Bozzo, A., Ades, M., Huijnen, V., Benedetti, A., Engelen, R., Peuch, V.-H., and Morcrette, J.-J.: Description and evaluation of the tropospheric aerosol scheme in the European Centre for Medium-Range Weather Forecasts (ECMWF) Integrated Forecasting System (IFS-AER, cycle 45R1), Geoscientific Model Development, 12, 4627–4659, https://doi.org/10.5194/gmd-12-4627-2019, 2019

---

## Author Comment (AC2) · 18 Sep 2020

We would like to thank Reviewer #3 for her/his time devoted and the constructive and helpful comments to which we will respond point by point.

The authors analyse a remarkable event of combined desert dust and biomass burning aerosol transport over Europe using observations and model results, and in the process evaluate some aspects of the CAMS air quality forecast system. Studying such events is important because of their quite drastic impact on the European air quality, potential implications on clouds and weather, and impairment of solar energy yield. Their distinct air quality signatures are good benchmarks for forecast systems. The subject is relevant and within the scope of ACP (there is some overlap with the GMD scope regarding the model evaluation aspect), the article is well organised and written so that I recommend publication after addressing the following comments:
We thank the Reviewer for the general comment.

Page 1, line 11: Throughout the manuscript you use the correlation (coefficient) r, here you present the shared variance (and I assume you are referring to r^2). Because the term is less common or even used with different meaning, and to be consistent with the main text, I suggest to stick to r.
We agree with the comment and thus we have replaced **"shared variance of 60%"** with **"correlation coefficient of 0.77"** in the Revised Manuscript (RM) P1, L11.

Page 1, line 13: Please expand IFS here.
Done.

Section 2.1: Some more information about the aerosol representation in the model would be helpful: does the model assume the different aerosol types to be externally mixed, both regarding the aerosol optical properties and regarding any chemical or physical interactions between different components? This would be interesting to know especially since the event under consideration involves dust, biomass burning aerosol and sea salt.
We agree with the comment raised by the Reviewer. Indeed, the IFS aerosol types are treated as externally mixed (separate particles). The following has been included in the RM P4, L7: **"The different IFS aerosol types are treated as externally mixed (Inness et al., 2019a)."**

Page 4, line 8 and 9: The units should be micro metres not metres and it has to be mentioned that the numbers represent radii not diameters. What bins are used for the other aerosol types, e.g., what is BC1 vs. BC2?
We thank the Reviewer for the comment, to provide a more explanatory description. We have replaced **"size bins"** with **"radius size bins"** (m were already replaced with μm in the discussion version of the manuscript). The two bins used for organic matter and black carbon aerosols stand for hydrophobic and hydrophilic. We have modified the respective discussion in the RM P4, L15-18 as follows: **"where ρ the air density, SS1 the sea salt radius size bin 1 (0.03-0.5 μm), SS2 the**

**sea salt radius size bin 2 (0.5-5 μm), DD1 the desert dust radius size bin 1 (0.03–0.55 μm), DD2 the desert dust radius size bin 2 (0.55–0.9 μm), DD3 the desert dust radius size bin 3 (0.9–20 μm), OM1 the hydrophobic organic matter, OM2 the hydrophilic organic matter, BC1 the hydrophobic black carbon, BC2 the hydrophilic black carbon and SU1 the aerosol sulfate (ECMWF, 2020).**

Page 4, Eqs. (1) and (2): I understand these equations follow recommendations elsewhere, but could you indicate in the text where the factors come from? Also, SS3 is not considered for PM2.5 (and PM10), because the radius is > 5 um, but according to the intervals provided above, not considering DD3 for PM2.5 seems to ignore dust particles between 0.9 and 1.25 um radius.

The factors applied in sea salt bins are used to transform sea salt from 80% relative humidity ambient conditions to dry, while the rest correspond to the fraction of each aerosol type included in $PM_{10}$ and $PM_{2.5}$. More information on the description and evaluation of the aerosol scheme used in IFS is provided by Remy et al. (2019) which we have included in the RM P4, L18-19 as follows: **"A detailed description and evaluation of the aerosol scheme used in IFS can be found in Remy et al. (2019)."** As for the disregarded DD3 radius size range 0.9-1.25 μm in $PM_{2.5}$, this is a limitation of the definition.

Page 5, line 10: Please expand LT once
Done. See P5, L16 of the RM.

Page 5, lines 28ff: Which selection criteria for the stations where used exactly?
Since here we examine the aerosol transport during Ophelia and the associated impacts on air quality in CAMS forecast systems, we only consider rural EMEP stations that fulfill the following criteria:
1. They are located over western Europe and away from the aerosol sources (i.e. Northern Africa and Iberian Peninsula).
2. They lie across the plumes of high AOD loadings during the examined event.
3. They exhibit remarkable increases in $PM_{10}$ and $PM_{2.5}$ surface concentrations (visual inspection).
Accordingly, we have replaced the sentence **"The stations are located over the broader western European areas where dust and biomass burning transport occured"** with the following in the RM P6, L2-4: **"The stations are located over western Europe and away from the dust and biomass burning sources, lie across the plumes of high AOD loadings, exhibiting significant increases in $PM_{10}$ and $PM_{2.5}$ surface concentrations during the examined event."**

Page 7, line 26: The FGE deserves to be introduced by an equation, moreover uppercase is more common.
We have added a new subsection (2.4 Statistical metrics) in the RM, where we present basic information and formulae for the Pearson correlation coefficient and the fractional gross error. We have also replaced all instances of fge with FGE following the suggestion in both text and figures of the RM.

Page 7, lines 30 to 34: How can this conclusion be aligned with Fig. 4, where in many regions with low AOD the AOD is enhance by data assimilation?

What we mean is that generally over areas with low "observed" AOD the IFS forecast tends to overestimate and vice versa. This agrees with the comment raised by the Reviewer, as over several areas with low observed AOD the IFS (with DA) forecast mostly overestimates. We have replaced **"low AOD values"** with **"low observed AOD values"** in the RM (P8, L26) for clarity. In support of the above we present here the differences between IFS $AOD_{550}$ and MODIS/Terra and Aqua $AOD_{550}$, which along with the left column of Figure 4 (MODIS/Terra and Aqua $AOD_{550}$ fields) confirms our conclusion.

[Figure]

Figs. 9 and 10: I find the similarity of the PM10 and PM2.5 composition surprising, normally I would expect a higher dust and sea salt fraction in PM10 than in PM2.5. Does this indicate some

limitation of the model? After all, from Eqs. (1) and (2) it is clear that the contributions in Fig. 9 and 10 cannot be that different, and the largest particles (SS3 and DD3) are not relevant at all, but is that realistic? Do the stations provide the PM2.5 and/or PM10 composition for comparison, or is there any other suitable data source to validate this aspect?

Indeed, as depicted from the $PM_{10}$ and $PM_{2.5}$ equations the contributions are similar. This was also the case in some sensitivity calculations we performed using random values for each aerosol type and bin. The equations themselves may play a role in that. The applied formulae are mostly empirical and thus we agree that may insert uncertainties (e.g. large particles are underrepresented). However, it should be also considered that in enhanced dust and sea salt conditions the total $PM_{10}$ and $PM_{2.5}$ concentrations are also higher likely resulting in similar relative contributions in $PM_{10}$ and $PM_{2.5}$. Unfortunately, from the examined stations only GB0048R and GB1055R EMEP stations provide fragments of $PM_{10}$ and $PM_{2.5}$ composition data for the examined period. These data are not sufficient for a comprehensive estimation of dust and sea salt contribution to $PM_{10}$ and $PM_{2.5}$ due to missing data and lack of all $PM_{10}$ and $PM_{2.5}$ aerosol components (available with gaps: $NH4^+$, $Ca^{++}$, $Cl^-$, $Mg^{++}$, $NO3^-$, $K^+$, $Na^+$ and $SO4^-$). From data for the GB1055R station during the 16-17 October PM peak, it seems that although the individual dust and sea salt component concentrations are higher in $PM_{10}$ compared to $PM_{2.5}$, it is the total $PM_{10}$ concentrations that are also higher compared to total $PM_{2.5}$ that probably result in similar contributions.

Generally, I believe links in footnotes and within the text are supposed to be be moved to the References section to comply with the journal standards.

Done. In the RM we have replaced all footnotes and links with References according to the Journal standards.

**References**

Inness, A., Ades, M., Agustí-Panareda, A., Barré, J., Benedictow, A., Blechschmidt, A.-M., Dominguez, J. J., Engelen, R., Eskes, H., Flemming, J., Huijnen, V., Jones, L., Kipling, Z., Massart, S., Parrington, M., Peuch, V.-H., Razinger, M., Remy, S., Schulz,M., and Suttie, M.: The CAMS reanalysis of atmospheric composition, Atmospheric Chemistry and Physics, 19, 3515–3556,https://doi.org/10.5194/acp-19-3515-2019, 2019

Osborne, M., Malavelle, F. F., Adam, M., Buxmann, J., Sugier, J., Marenco, F., and Haywood, J.: Saharan dust and biomass burning aerosols during ex-hurricane Ophelia: observations from the new UK lidar and sun-photometer network, Atmos. Chem. Phys., 19, 3557–3578, https://doi.org/10.5194/acp-19-3557-2019, 2019

Rémy, S., Kipling, Z., Flemming, J., Boucher, O., Nabat, P., Michou, M., Bozzo, A., Ades, M., Huijnen, V., Benedetti, A., Engelen, R., Peuch, V.-H., and Morcrette, J.-J.: Description and evaluation of the tropospheric aerosol scheme in the European Centre for Medium-Range Weather Forecasts (ECMWF) Integrated Forecasting System (IFS-AER, cycle 45R1), Geoscientific Model Development, 12, 4627–4659, https://doi.org/10.5194/gmd-12-4627-2019, 2019

---

## Author Comment (AC3) · 18 Sep 2020

We would like to thank Reviewer #2 for her/his time devoted and the constructive and helpful comments to which we will respond point by point.

The authors analyse an interesting meteorological/aerosol event named Ophelia, that took place in mid-October 2017 over western Europe. They use CAMS global day-1 forecast data to assess the aerosol spatiotemporal distribution and determine the source and type of aerosol species. Validating the performance of CAMS in these kind of events is quite crucial to understand the current limitations in CAMS aerosol forecast system and the possible upgrades that need to be planned (e.g. assimilating more and different kinds of aerosol observations). The manuscript is well written providing all the necessary information to introduce the event to the reader. I would recommend for publication in ACP after addressing some minor comments below:

We thank the Reviewer for the general comment.

CAMS AOD is evaluated using dependent observations (as mentioned at P7, L10-11) from MODIS Dark Target and Deep Blue algorithms showing reasonable agreement with the observations and improvement in both correlation and bias in comparison to the no assimilation experiment. The paper does not provide a comparison with independent observations to prove that data assimilation improves AOD. I would suggest to mention again in conclusions that this is a dependent evaluation (P10, L28-31) or perform a small analysis using independent observations (e.g. AERONET AOD) to prove that CAMS AOD really improves, although the latter option may be out of the scope of this paper.

We agree with the comment raised from the Reviewer and thus in the Conclusions of the RM (P12, L8-9) we explicitly state that findings are drawn from a depended evaluation: **"The dependent evaluation against MODIS satellite observations reveals a satisfactory agreement with CAMS global AOD$_{550}$ (R=0.77 and FGE=0.4), while the comparison.."**. As assumed by the Reviewer, a comparison with AERONET AOD was not a primary scope of the present paper. Moreover, there is a limited availability of AERONET data during the passage of Storm Ophelia (due to cloud presence), with a few stations exhibiting fragments of observations (see also Figure 9 in the study of Osborne et al. (2019)), which in any case are not sufficient to conduct a comprehensive independent evaluation of CAMS global AOD product taking also into account its temporal resolution (3-hour).

Figure 6 and discussion starting at P8, L1: A very interesting analysis highlighting the contribution of each species to AOD. Nevertheless the contribution of each species to AOD may differ depending on the aerosol-species optical properties. Black carbon AOD is low (and Absorption AOD high) in comparison to the other species. A 15% to 25% contribution of black carbon AOD to the total AOD might seem mediocre, but climatologically is only observed during fire season in the Tropical band. Maybe the authors would like to comment on that.

We agree with the comment and therefore we include the following sentence in the RM (P9, L3-5): **"Such black carbon contributions are considered high being similar to climatological contributions over global fire hot spots (e.g. summertime central southern Africa; Penning de Vries et al. (2015))."**

Figure 7 and discussion starting at P8, L17: Although this is more of a qualitative comparison to discuss the vertical distribution of aerosols per species, the depiction of CALIPSO and IFS are very different above 4km in both tracks. Is it possible to conclude something about the observations or the model? For example is CALIPSO unable to retrieve trustworthy measurements during the pass of Ophelia above 4km or is IFS overestimating aerosol mixing ratio above a certain height? The respective Figure has been revised correcting a minor bug in our code and converting IFS sea salt from 80% RH ambient conditions to dry, as suggested by ECMWF, which was inadvertently omitted. However, the Figure and the respective discussion have not changed substantially. Regarding the inconsistencies above 4km, this is a rather complex and still uncertain issue. As discussed comprehensively in Georgoulias et al. (2018) the non-zero aerosol concentrations appearing systematically at heights above 5 km in model simulations compared to remote sensing observations (also see Mona et al., 2014; Binietoglou et al., 2015; Cuevas et al., 2015 and Ansmann et al., 2017) could be due to:
- The way the model deals with the aerosol distribution in different size bins and aerosol deposition, vertical transport and mixing.
- The IFS assimilation process and the fact that nitrate aerosols are missing in the present IFS aerosol version, which might lead to the appearance of small aerosol concentrations further high.
- The limitation of CALIPSO in detecting aerosol layers with signals lower than the satellite's signal-to-noise ratio.

**References**

Ansmann, A., Rittmeister, F., Engelmann, R., Basart, S., Jorba, O., Spyrou, C., Remy, S., Skupin, A., Baars, H., Seifert, P., Senf, F., and Kanitz, T.: Profiling of Saharan dust from the Caribbean to western Africa – Part 2: Shipborne lidar measurements versus forecasts, Atmos. Chem. Phys., 17, 14987–15006, https://doi.org/10.5194/acp-17-14987-2017, 2017

Binietoglou, I., Basart, S., Alados-Arboledas, L., Amiridis, V., Argyrouli, A., Baars, H., Baldasano, J. M., Balis, D., Belegante, L., Bravo-Aranda, J. A., Burlizzi, P., Carrasco, V., Chaikovsky, A., Comerón, A., D'Amico, G., Filioglou, M., Granados-Muñoz, M. J., Guerrero-Rascado, J. L., Ilic, L., Kokkalis, P., Maurizi, A., Mona, L., Monti, F., Muñoz-Porcar, C., Nicolae, D., Papayannis, A., Pappalardo, G., Pejanovic, G., Pereira, S. N., Perrone, M. R., Pietruczuk, A., Posyniak, M., Rocadenbosch, F., Rodríguez-Gómez, A., Sicard, M., Siomos, N., Szkop, A., Terradellas, E., Tsekeri, A., Vukovic, A., Wandinger, U., and Wagner, J.: A methodology for investigating dust model performance using synergistic EARLINET/AERONET dust concentration retrievals, Atmos. Meas. Tech., 8, 3577–3600, https://doi.org/10.5194/amt-8-3577-2015, 2015

Cuevas, E., Camino, C., Benedetti, A., Basart, S., Terradellas, E., Baldasano, J. M., Morcrette, J. J., Marticorena, B., Goloub, P., Mortier, A., Berjón, A., Hernández, Y., Gil-Ojeda, M., and Schulz, M.: The MACC-II 2007–2008 reanalysis:

atmospheric dust evaluation and characterization over northern Africa and the Middle East, Atmos. Chem. Phys., 15, 3991–4024, https://doi.org/10.5194/acp-15-3991-2015, 2015

Georgoulias, A. K., Tsikerdekis, A., Amiridis, V., Marinou, E., Benedetti, A., Zanis, P., Alexandri, G., Mona, L., Kourtidis, K. A., and Lelieveld, J.: A 3-D evaluation of the MACC reanalysis dust product over Europe, northern Africa and Middle East using CALIOP/CALIPSO dust satellite observations, Atmos. Chem. Phys., 18, 8601–8620, https://doi.org/10.5194/acp-18-8601-2018, 2018

Mona, L., Papagiannopoulos, N., Basart, S., Baldasano, J., Binietoglou, I., Cornacchia, C., and Pappalardo, G.: EARLINET dust observations vs. BSC-DREAM8b modeled profiles: 12-year-long systematic comparison at Potenza, Italy, Atmos. Chem. Phys., 14, 8781–8793, https://doi.org/10.5194/acp-14-8781-2014, 2014

Osborne, M., Malavelle, F. F., Adam, M., Buxmann, J., Sugier, J., Marenco, F., and Haywood, J.: Saharan dust and biomass burning aerosols during ex-hurricane Ophelia: observations from the new UK lidar and sun-photometer network, Atmos. Chem. Phys., 19, 3557–3578, https://doi.org/10.5194/acp-19-3557-2019, 2019

Penning de Vries, M. J. M., Beirle, S., Hörmann, C., Kaiser, J. W., Stammes, P., Tilstra, L. G., Tuinder, O. N. E., and Wagner, T.: A global aerosol classification algorithm incorporating multiple satellite data sets of aerosol and trace gas abundances, Atmos. Chem. Phys., 15, 10597–10618, https://doi.org/10.5194/acp-15-10597-2015, 2015.